# Massive Southern Ocean phytoplankton bloom fed by iron of possible hydrothermal origin

Casey M. S. Schine [1✉], Anne-Carlijn Alderkamp[2], Gert van Dijken [1], Loes J. A. Gerringa[3], Sara Sergi[4], Patrick Laan[3], Hans van Haren[3], Willem H. van de Poll [5] & Kevin R. Arrigo [1]

Primary production in the Southern Ocean (SO) is limited by iron availability. Hydrothermal vents have been identified as a potentially important source of iron to SO surface waters. Here we identify a recurring phytoplankton bloom in the high-nutrient, low-chlorophyll waters of the Antarctic Circumpolar Current in the Pacific sector of the SO, that we argue is fed by iron of hydrothermal origin. In January 2014 the bloom covered an area of ~266,000 km$^2$ with depth-integrated chlorophyll $a > 300$ mg m$^{-2}$, primary production rates >1 g C m$^{-2}$ d$^{-1}$, and a mean $CO_2$ flux of $-0.38$ g C m$^{-2}$ d$^{-1}$. The elevated iron supporting this bloom is likely of hydrothermal origin based on the recurrent position of the bloom relative to two active hydrothermal vent fields along the Australian Antarctic Ridge and the association of the elevated iron with a distinct water mass characteristic of a nonbuoyant hydrothermal vent plume.

[1] Department of Earth System Science, Stanford University, Stanford, CA 94305, USA. [2] Biology Department, Foothill College, Los Altos Hills, CA 94022, USA. [3] Royal Netherlands Institute for Sea Research (NIOZ), PO Box 59, 1790 AB Den Burg, the Netherlands. [4] Sorbonne Université, CNRS, IRD, MNHN, Laboratoire d'Océanographie et du Climat: Expérimentations et Approches Numériques (LOCEAN-IPSL), F-75005 Paris, France. [5] Department of Ocean Ecosystems, Energy, and Sustainability Research Institute Gronigen, Faculty of Science and Engineering, University of Groningen, Nijenborgh 7, 9747 AG Groningen, the Netherlands. ✉email: cmsmith9@stanford.edu

Primary productivity in the Southern Ocean (SO) is critical in governing atmospheric $CO_2$ levels[1]. Because net primary productivity (NPP) in the SO is limited by the availability of iron (Fe)[2–4], understanding the distribution of Fe availability in the SO and its relationship to spatial patterns in NPP is essential to quantify the capacity of the SO to act as a carbon sink.

In 2010, Tagliabue et al.[5] proposed that Fe of hydrothermal origin is an essential part of the SO Fe budget and that the observed Fe distribution in the SO cannot be replicated without the inclusion of hydrothermal sources of Fe. Tagliabue et al.[5] further proposed that the utilization of hydrothermal Fe by phytoplankton fuels increased NPP in the SO that results in 5–30% more carbon export. The first report of hydrothermal vent emissions fueling a large phytoplankton bloom was from the Indian sector of the SO around 40°E near the Southwest Indian Ridge (SWIR). Ardyna et al.[6] used historical measurements of elevated $\delta^3$He to suggest that deep water concentrations of Fe in the vicinity of the bloom (observed by BGC-Argo floats) were elevated and of hydrothermal origin. They further demonstrated that enhanced eddy kinetic energy (EKE) resulting from the interaction of currents with bottom topography brought the hydrothermal vent emissions to the surface, stimulating phytoplankton growth.

Two active hydrothermal vent fields were recently found along the Australian Antarctic Ridge (AAR; Fig. 1), a series of ridge segments and transform faults in the Pacific Sector of the SO[7]. KR1, the southerly of the two vent systems (KR2 is further to the northwest), coincides with the positions of the southern boundary of the Antarctic Circumpolar Current (sbACC) and the southern ACC front (sACCf), which run very close together in this part of the ACC (Fig. 1a). The position of KR1 is also coincident with a large area of perennially elevated chlorophyll a (Chl a) and NPP (Fig. 1b) visible in satellite-based climatologies extending back to 1978[8–12]. This indicates that a phytoplankton bloom develops in the same location near KR1 almost every year.

Here we describe observations from a SO research cruise that, for the first time, sampled surface waters above the AAR in the region of perennially elevated NPP near KR1. During that cruise, we measured the hydrographic conditions, seawater chemistry, and biological responses associated with a massive phytoplankton bloom in the otherwise high-nutrient, low-chlorophyll (HNLC) waters of the ACC. By combining field data and satellite imagery, we were able to shed light on the likely cause for such an intense phytoplankton bloom in a region of generally low NPP.

## Results

**Characteristics of the AAR bloom.** Satellite imagery showed that the position of the bloom, adjacent to the AAR and the sbACC (Fig. 1c), to the northwest of the Ross Sea (Fig. 1d) (hereafter referred to as the AAR bloom), remained relatively stationary from December 2013 through February 2014 and did not track the receding ice edge (Supplementary Fig. 1). From ocean color data, we calculated that the area of the AAR bloom where the mean Chl a concentration from November through February exceeded 0.25 mg m$^{-3}$ was ~266,000 km$^2$.

Our in situ measurements confirmed that the AAR bloom observed in satellite imagery was characterized by elevated phytoplankton biomass and NPP. Our underway data showed an increase in fluorescence (Supplementary Fig. 1) associated with the AAR bloom identified in satellite images. Chl a concentration within the top 40 m of the water column (Fig. 2a) ranged from 5.00–7.09 mg m$^{-3}$ inside the bloom (stations 130/151, 131, and 150) and only 0.33–0.97 mg m$^{-3}$ outside the bloom (stations 119 and 140/149). The depth-integrated Chl a ranged from 56.7 mg m$^{-2}$ at station 119 and 60.0 mg m$^{-2}$ at station 140 outside of the

bloom to 114.8 mg m$^{-2}$ at station 135 on the edge of the bloom to 319.0 mg m$^{-2}$ at station 130 and 255.0 mg m$^{-2}$ at station 131, in the interior of the bloom. Five stations (130/151, 131, and 150 inside the bloom as well as 120 and 133 on the edge of the bloom) had depth-integrated Chl a exceeding 140 mg m$^{-2}$. Particulate organic carbon (POC) concentrations inside the bloom ranged from 705.4 to 833.9 mg m$^{-3}$, with values of only 101.2–189.9 mg m$^{-3}$ outside the bloom (Fig. 2b).

Depth-integrated NPP inside the AAR bloom was around three times higher than outside of the bloom. At stations 130 and 150 where Chl a and POC concentrations were high, NPP reached 1.14 and 1.10 g C m$^{-2}$ d$^{-1}$, respectively. In contrast, at stations with low Chl a and POC (stations 119 and 140) depth-integrated NPP was only 0.40 and 0.35 g C m$^{-2}$ d$^{-1}$, respectively.

The AAR bloom was also associated with significant $CO_2$ and nutrient drawdown by phytoplankton. Surface $p$$CO_2$ outside the bloom ranged from 394 to 410 µatm while $p$$CO_2$ inside the bloom was reduced to 196–228 µatm (Fig. 1d). Similarly, nitrate concentrations inside the bloom ranged from 13.52 to 18.86 µM in the top 30 m, much lower than near-surface concentrations outside of the bloom of 27.03–29.62 µM and subsurface concentrations below 40 m of 30.45–32.73 µM at stations both inside and outside of the bloom (Fig. 2c). The depth of the mixed layer (MLD) showed a wide range across all sampling stations and we found no relationship between MLD and Chl a or POC concentration in the upper mixed layer. MLD at stations inside the bloom (stations 130/151, 131, and 150) ranged from 23 to 30 m, outside the bloom (stations 119 and 140/149) from 16 to 35 m, and on the margin of the bloom (stations 120, 133,135, and 136) from 17 to 41 m.

CHEMTAX analysis of HPLC pigments (confirmed by microscopy) showed that the AAR bloom was dominated by the haptophyte *Phaeocystis antarctica*, which accounted for 90–100% ($n = 9$) of the phytoplankton Chl a inside the bloom. Outside the bloom, diatoms were dominant, comprising 61.6–90.4% ($n = 11$) of phytoplankton Chl a. The dominance of *P. antarctica* inside the bloom was also strongly indicated by the lack of silicate drawdown, despite high nitrate drawdown, with silicate concentrations inside the bloom (65.10–67.89 µM) roughly equivalent to those outside the bloom (60.15–71.88 µM).

Dissolved Fe (DFe) was substantially depleted in surface waters both inside and outside the AAR bloom, with concentrations falling to 0.04–0.05 nM (Fig. 2d). The substantial depletion of DFe in surface waters at bloom stations combined with the incomplete drawdown of nitrate indicates that DFe was limiting the productivity of the bloom at the time of our sampling. However, mean subsurface DFe (Fig. 3a) concentrations measured just below the ferricline (from 90 to 250 m) at bloom stations (0.29–0.35 nM at 150 m) were significantly higher ($p < 0.0001$) than DFe concentrations measured below the ferricline at stations outside the bloom (0.13–0.16 nM at 150 m). Deep (>250 m) DFe concentrations inside the bloom were also significantly elevated relative to those outside the bloom down to our deepest sampling depth of 2000 m ($p < 0.0001$). Vertical diffusivity (log-transformed to comply with normality assumptions) was significantly higher ($p = 0.0002$) in subsurface (90–250 m) waters beneath the bloom (mean = $2.8 \times 10^{-3}$ m$^{-2}$ s$^{-1}$) than outside the bloom (mean = $0.5 \times 10^{-3}$ m$^{-2}$ s$^{-1}$; Fig. 3b).

Additionally, profiles of DFe (Fig. 3d), potential temperature (Fig. 3e), salinity (Fig. 3f), and oxygen (Fig. 3g) plotted against density reveal that elevated subsurface DFe concentrations are associated with an anomalous water mass located within the potential density anomaly range of 27.6–27.8 kg m$^{-3}$. At bloom stations, waters with a potential density anomaly between 27.6 and 27.8 kg m$^{-3}$ were colder, fresher, and more oxygenated than at non-bloom stations.

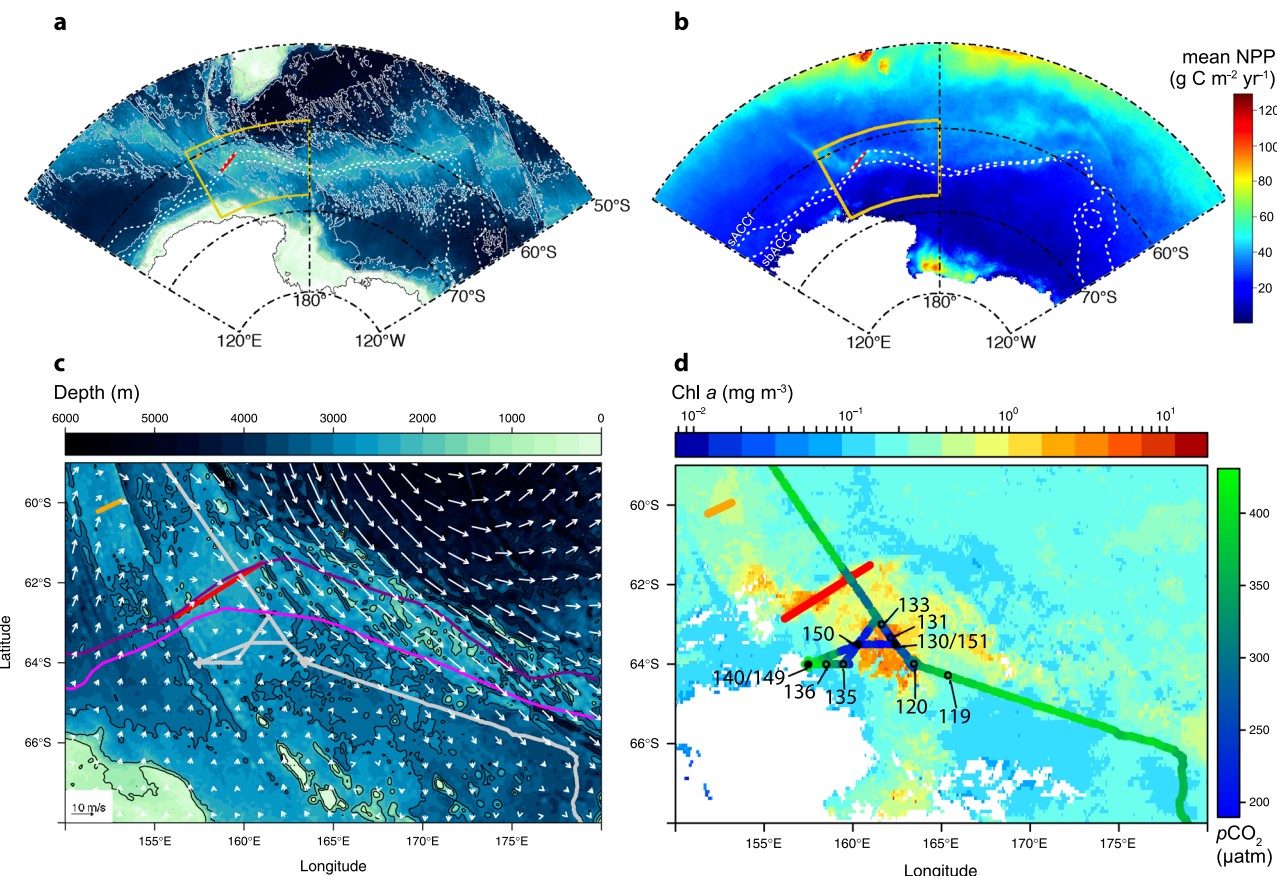

**Fig. 1 Bathymetry and phytoplankton distribution in the Pacific sector of the Southern Ocean.** On all figures, red and orange lines indicate the positions of the ridges KR1 and KR2, respectively[7]. The positions of southern Antarctic Circumpolar Current (ACC) front (sACCf) and the more southerly positioned southern boundary of the ACC (sbACC)[35], are shown by the white dashed lines in **a** and **b** and by the pink lines in **c**. **a** Map of Pacific sector bathymetry, with yellow inset box that corresponds to the areas shown in **c** and **d**. **b** Map of Pacific sector climatological net primary production (NPP; 1997–2019), indicating the return of the bloom to the same location most years. **c** Map of bathymetry and January 2014 geostrophic currents from Dotto et al.[14]. Cruise track is shown as solid gray line. **d** Map of mean satellite chlorophyll *a* (Chl *a*) from November 2013-February 2014 with underway $p$CO$_2$ overlaid along the cruise track. Stations are indicated by black circles and labeled with a station number. Closed circles indicate stations where a cast to 2000 m was conducted to sample dissolved Fe (DFe). The distance from the center of KR1 to station 150 is 168 km. Bathymetry data are from https://www.ngdc.noaa.gov/mgg/global/. White areas indicate land in **a** and **b** and an absence of valid Chl *a* data in **d** which is either land, sea-ice, or persistent cloud cover. Note that the satellite Chl *a* image that covers the bloom period **d** is a composite of multiple months and the location of the bloom in this image may not exactly match our in situ measurements. The convergence zone visible to the south of our study area **c** falls within the boundaries of the western side of the Ross Gyre. However, the location of this zone shifts substantially on a monthly timescale and is unlikely to impact the bloom.

**DFe supply**. To determine if the observed drawdown of DFe in surface waters of the AAR bloom was sufficient to support the accumulated phytoplankton biomass, we calculated the ratio of DFe removed from surface waters (assuming the depletion in DFe concentration between the surface and 150–200 m depth was due to phytoplankton uptake) to POC accumulated (the average POC concentration above 40 m). At stations within the bloom, the ratio of DFe removed to POC accumulated ranged from 4.0 to 5.2 $\mu$mol mol$^{-1}$, which falls within the range of cellular Fe:C ratios reported for *P. antarctica* (2.3–8.6 $\mu$mol mol$^{-1}$)[13], the dominant phytoplankton inside the bloom. At stations outside of the bloom, the ratio of DFe removed to POC accumulated ranged from 13.0 to 21.2 $\mu$mol mol$^{-1}$, which is consistent with the range of cellular Fe:C ratios for diatoms (10–20 $\mu$mol mol$^{-1}$)[13], the dominant phytoplankton group in these waters. These findings demonstrate that the amount of DFe removed from surface waters was sufficient to support the observed phytoplankton biomass accumulation both inside and outside of the bloom.

In order to assess whether the rate of CO$_2$ fixation (i.e., NPP) by phytoplankton measured in situ could be supported by the delivery of DFe across the ferricline, we compared the calculated vertical DFe flux to depth-integrated NPP for stations 130 and 150 inside the bloom and stations 119 and 140 outside of the bloom. The ratio of DFe delivered to CO$_2$ fixed inside the bloom was 4.37–4.70 $\mu$mol:mol, consistent with the cellular Fe:POC range for *P. antarctica*[13]. Outside the bloom, the DFe delivery to CO$_2$ fixation ratio was 2.82–3.14 $\mu$mol:mol, which is much lower than the cellular Fe:POC range for diatoms[13], suggesting that the community of diatoms growing outside of the bloom would be extremely Fe-stressed. Presumably, *P. antarctica* was the dominant phytoplankton in the AAR bloom due to its favorable Fe requirements.

We also examined the relationship between depth-integrated Chl *a* and vertical DFe flux using linear regression analysis. Depth-integrated Chl *a* was significantly correlated with vertical DFe flux such that higher DFe flux was associated with higher depth-integrated Chl *a* (Fig. 3c; $p = 0.04$; $r^2 = 0.45$). Depth-integrated Chl *a* was not significantly correlated with either vertical diffusivity ($p = 0.18$) or the DFe concentration gradient across the ferricline ($p = 0.14$).

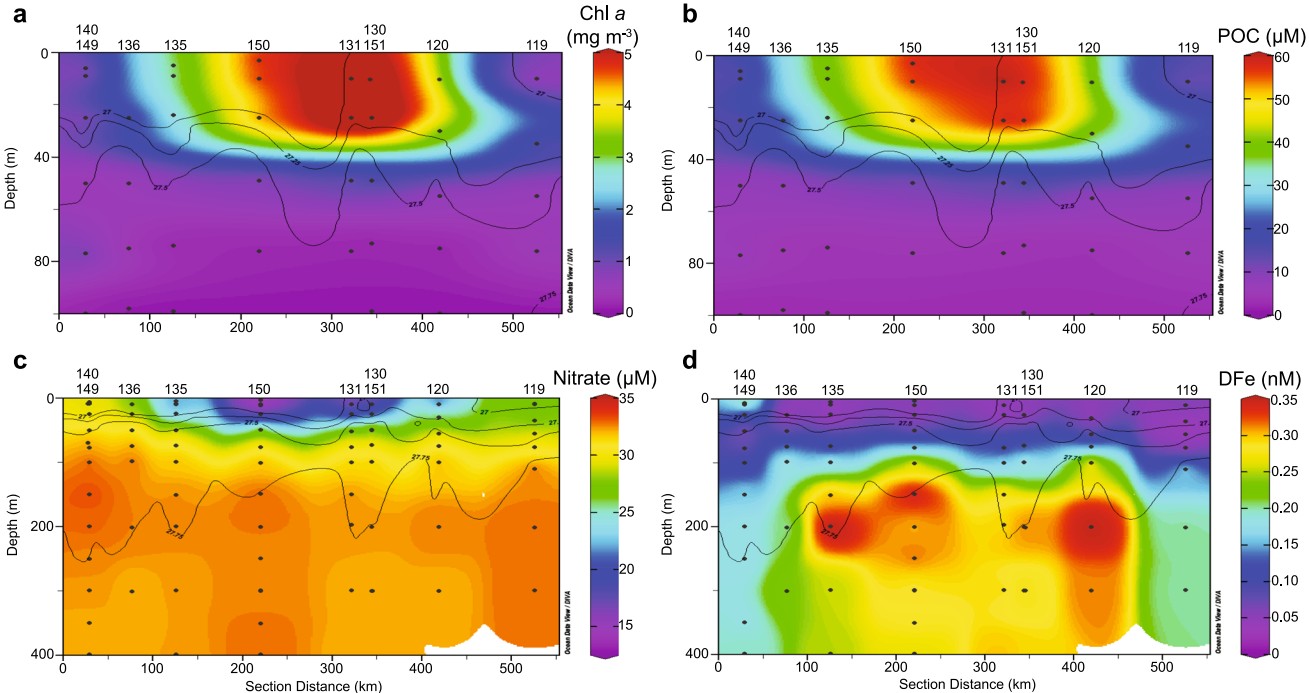

**Fig. 2 Biomass and nutrients in the water column in and around the bloom.** Vertical sections of **a** chlorophyll *a* (Chl *a*), **b** particulate organic carbon (POC), **c** nitrate, and **d** dissolved Fe (DFe) from in situ measurements. Note that the depth in panels **a** and **b** extends only to 100 m, while the depth in panels **c** and **d** extends to 400 m. Station numbers are listed above each section, block dots indicate sampling depths, and black lines show isopycnals. A map of the portion of the cruise shown in these sections and the stations that correspond with the stations here is shown in Fig. 1d. Station 133 is on the map but not included in the section.

**Hydrologic context**. To better understand the hydrologic conditions influencing the AAR bloom, we examined the NPP, deep EKE, isopycnal depth, and bathymetry of the waters in and around the bloom (Supplementary Fig. 2), looking specifically at the way these variables change along the sACCf between 100°E and 180°E (Fig. 4). Along the sACCf, the shallower bathymetry of KR1 begins around 156°E, although isopycnals reach their minimum depth upstream (~150°E) of this shift in bathymetry. Deep EKE (at ~1000 m) in the region of the AAR is low, falling below 10 cm$^2$ s$^{-2}$ between 150°E and 160°E. NPP along the sACCf climbs steadily starting around 153°E, reaching a maximum near 160°E downstream of where the sACCf passes over KR1.

A Lagrangian particle tracking climatology (1997–2019; Supplementary Fig. 3) shows that particles released directly above KR1 follow prevailing surface currents to the west and that particles released above KR2 move both to the southeast and the northwest, consistent with the climatological shape of the bloom observed in our satellite NPP analysis (Fig. 1b). The currents in this region tend to be slow (Fig. 1c) and thus the residence time of water that has moved through the region above the vents is long (Supplementary Fig. 3). Geostrophic surface currents (calculated from satellite altimetry data[14]) were directed southeasterly during the month of January 2014 in the vicinity of the AAR bloom, decreasing in intensity from the northeast to the southwest across the bloom (Fig. 1c). The geostrophic surface currents in January 2014 are consistent with the results from the Lagrangian particle tracking climatology.

## Discussion

The AAR bloom that we sampled in the Pacific Sector of the SO was large, long-lived, and contained a substantial amount of phytoplankton biomass. Both climatological NPP (Fig. 1b) and

annual satellite Chl *a* imagery (Supplementary Fig. 4) show that the bloom has developed in the vicinity of KR1, KR2, or both vent ridges in 20 out of the last 22 years. During our cruise in January of 2014, the standing stock of Chl *a* in the AAR bloom (140–300 mg Chl *a* m$^{-2}$) exceeded the depth-integrated Chl *a* of the most intense ACC bloom thus far measured by BGC-Argo floats (98.1 mg Chl *a* m$^{-2}$)[6]. It also eclipsed the large Fe-enriched blooms located downstream of the relatively shallow Kerguelen (south of the Polar Front, which excludes the portion of the bloom directly downstream of the island) and Crozet plateaus, which have a mean phytoplankton biomass of ~120 mg Chl *a* m$^{-2}$ [15,16] and 180–229 mg Chl *a* m$^{-2}$ [17], respectively. The maximum depth-integrated Chl *a* measured in our bloom is more than three times higher than that measured in the hydrothermally-driven SWIR blooms[6].

The productivity rates in the AAR bloom were also very high compared with other blooms in the ACC. Our daily NPP measurements in the bloom (1.10–1.14 g C m$^{-2}$ d$^{-1}$) are 2.7-3.8 times higher than the satellite-based daily NPP estimates at the peak of the spring bloom in open SO waters (0.30–0.40 g C m$^{-2}$ d$^{-1}$)[11]. NPP in the AAR bloom was within the range of that measured over the Crozet Plateau (0.52–3.00 g C m$^{-2}$ d$^{-1}$ at stations where depth-integrated Chl *a* exceeded 50 mg m$^{-2}$)[17]. The areal extent of the AAR bloom (266,000 km$^2$) is also within the range for the Crozet Plateau bloom (70,500–355,000 km$^2$ from 1998 to 2007), although it was substantially smaller than the Kerguelen Plateau bloom (340,000–1,600,000 km$^2$ from 1998 to 2007)[18]. The Kerguelen bloom typically persists from November through January and the Crozet bloom from October through December[18], which is approximately equivalent to the length of the AAR bloom (December through February).

The high biomass and longevity of the AAR bloom is likely explained by the combination of elevated DFe concentrations in surface waters at the onset of the growing season as well as high

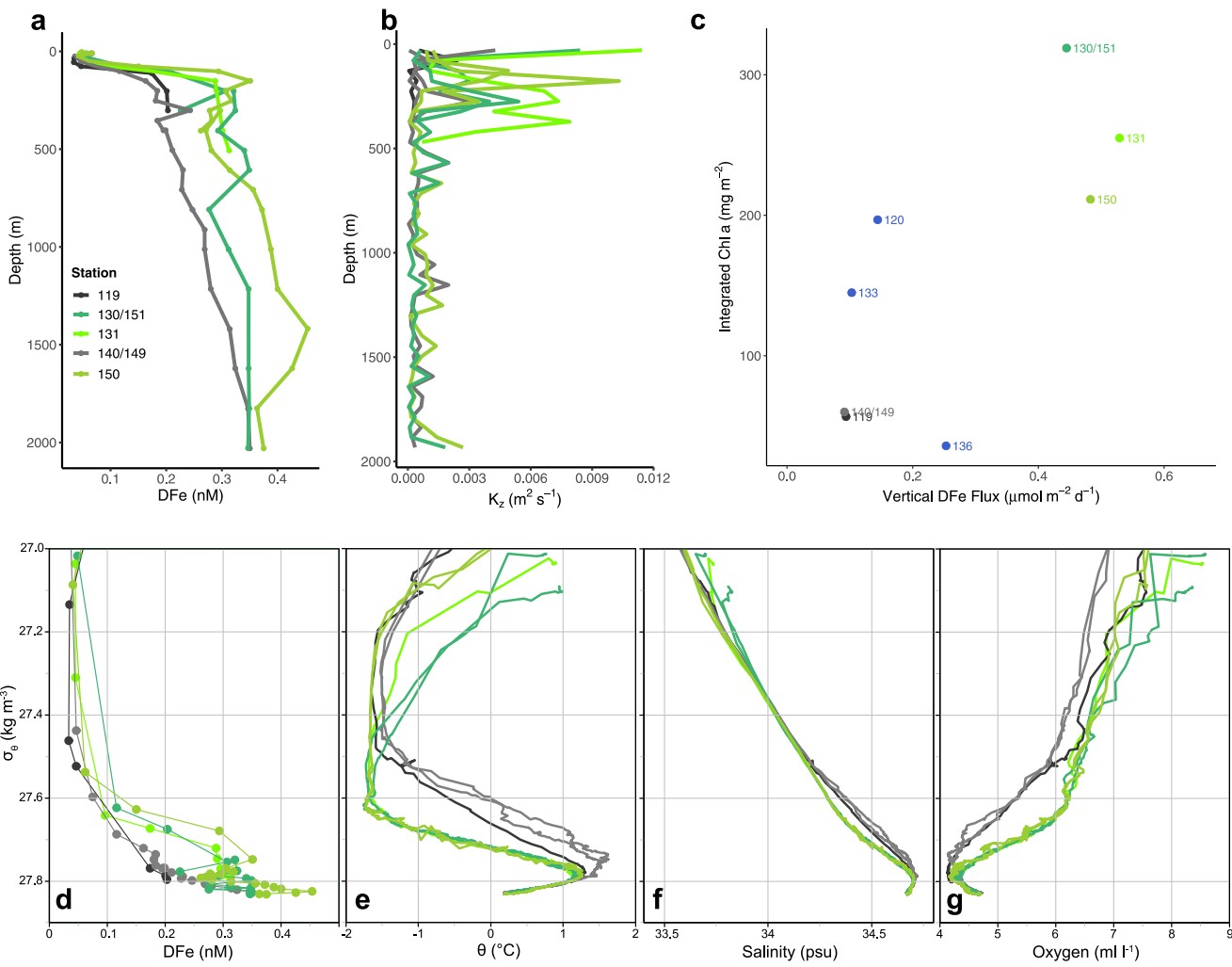

**Fig. 3 Water mass characteristics in the bloom versus outside of the bloom.** Depth profiles of **a** dissolved Fe (DFe) and **b** vertical diffusivity ($K_z$) for stations inside the bloom (green lines) and stations outside the bloom (gray lines). **c** Depth-integrated chlorophyll $a$ (Chl $a$) versus DFe flux. Density profiles of **d** DFe, **e** potential temperature, **f** salinity, and **g** oxygen concentration. We do not include error bars for our DFe measurements or our depth-integrated Chl $a$ measurements in either the depth or density profiles because the standard deviation for all DFe measurements was below 0.012 nM, and the error associated with all depth-integrated Chl $a$ measurements was below 0.12%. Stations shown in blue **c** fall on the edge of the bloom and are not included in the other plots.

rates of DFe delivery to the upper mixed layer through the austral summer, consistent with the bloom phenology proposed by Tagliabue et al.[19]. Higher rates of DFe delivery to surface waters inside the AAR bloom are driven by substantially greater DFe concentrations below the ferricline (0.31–0.35 nM) compared to that of non-bloom stations (0.12–0.17 nM) combined with a higher vertical diffusivity. While the vertical gradient of DFe measured at our bloom stations is only slightly higher than that reported for the Kerguelen Plateau[20], the vertical DFe fluxes we calculated (0.44–0.53 µmol m$^{-2}$ d$^{-1}$ at stations 130, 131, and 150; Fig. 3c) are substantially higher. Blain et al.[20] reported a vertical DFe flux at a high biomass station near the Southern Kerguelen Plateau of 0.03 µmol m$^{-2}$ d$^{-1}$ and Tagliabue et al.[19] estimated a vertical diffusive DFe flux range of 0.0016–0.0157 µmol m$^{-2}$ d$^{-1}$ for the SO. Our vertical DFe fluxes are an order of magnitude higher than these values.

The difference between our vertical DFe flux estimates and those reported previously[19,20] originates from the fact that our calculated vertical diffusivity ($K_z$) across the ferricline (0.9–4.4 × 10$^{-3}$ m$^{-2}$ s$^{-1}$) is an order of magnitude higher than the mean value generally reported for the SO of 10$^{-4}$ to 10$^{-5}$ m$^{-2}$ s$^{-1}$ [19–22].

Although our $K_z$ values are relatively high, they still fall within the range of values reported previously in the SO. $K_z$ values as high as 10$^{-1}$ have been reported in the upper mixed layer of the SO during times of strong winds[23,24], and values of 10$^{-3}$ to 10$^{-4}$ are typical of the seasonal pycnocline[23]. We observed that buoyancy frequency was slightly higher at bloom stations (though not significantly) at the same depths where elevated $K_z$ was observed (Supplementary Fig. 5) and that these elevated buoyancy frequency values extended deeper into the water column at bloom stations. Therefore, it is possible that the stronger stratification measured at blooms stations could have led to the breaking of surface-generated near-inertial waves, thereby generating the elevated vertical diffusivities we observed[25].

The potential sources of new Fe in the SO that could fuel the AAR bloom are aeolian dust deposition[26,27], iceberg[13,28] and sea ice melt[13,29,30], sediments (continental shelf advection)[31], upwelling of deep water[19,32], and hydrothermal vent emissions that recently have been shown to be a significant source of Fe to deep waters in the SO[5,32,33]. We can rule out aeolian dust flux and iceberg melt, as estimates of dust deposition south of 60°S are vanishingly small (0.001–0.003 mmol Fe m$^{-2}$ yr$^{-1}$)[26,27] and the

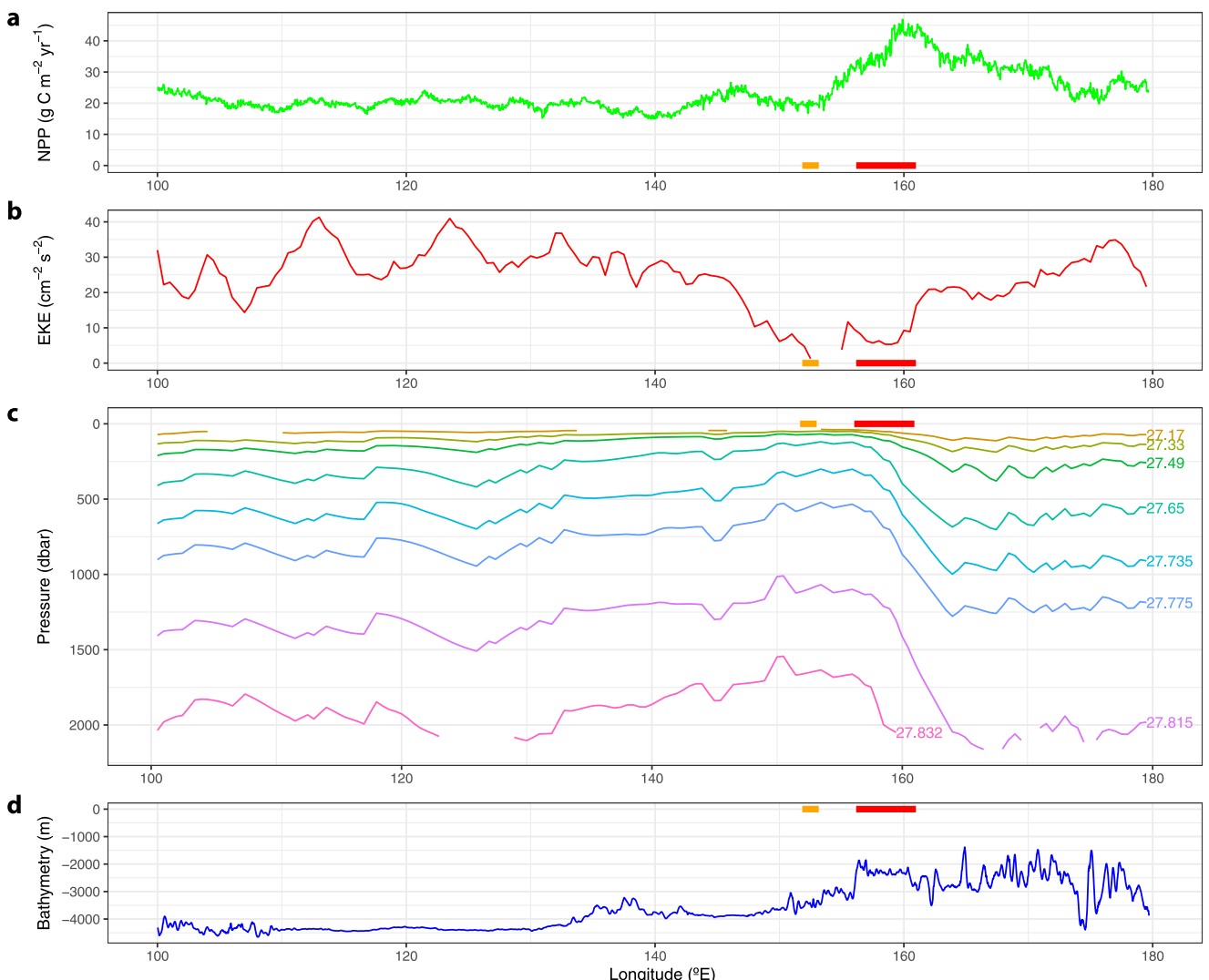

**Fig. 4 Position of the bloom relative to factors that promote upwelling along the southern Antarctic Circumpolar (ACC) front (sACCf). a** Net primary production (NPP), **b** eddy kinetic energy (EKE), **c** isopycnal pressure, and **d** bathymetry along the sACCf according to the front positions of Orsi et al.[35]. Supplementary Fig. 2 shows maps of the data used in this figure with the position of the sACCf. The longitudinal position of KR1 and KR2 are shown by the red line and orange line, respectively, on each panel. Bathymetry data are from https://www.ngdc.noaa.gov/mgg/global/.

path of icebergs in the Pacific sector of the SO lies too far south to impact the AAR bloom[28]. While sea ice may account for some DFe input[30], we can rule it out as the primary source of DFe since the bloom does not track the sea ice edge (Supplementary Fig. 1). Furthermore, the input of DFe from dust deposition, iceberg melt, and sea ice cannot account for the elevated DFe concentrations observed down to 2000 m. Advection from continental shelves is also not a strong candidate. While weak currents appear to move from the region of the shelf in the south into our study area, results from a Lagrangian model of horizontal DFe delivery shows almost no advected DFe from continental shelves making it to the vicinity of the AAR bloom[6,31]. Most notably, none of these potential Fe sources can explain why the bloom recurs in the same location every year. The remaining potential sources of DFe are upwelling of Upper Circumpolar Deep Water (UCDW) and hydrothermal vent emissions.

Upwelling is undoubtedly playing a major role in waters around the AAR bloom. The co-location of an upwelling front with a bathymetric feature such as a ridge, both of which are present at the AAR bloom site, has been shown to result in persistent blooms associated with upwelling fronts[10]. The position of the AAR bloom is adjacent to the sbACC and sACCf,

which flow very close together in this region. This means that the AAR bloom is positioned close to the steeply tilted and outcropping isopycnals associated with these fronts. In addition to the meridional shallowing of isopycnals southward across the sACCf and sbACC, there is also a zonal shallowing of isopycnals along the sACCf as the ACC flows over the AAR (Fig. 4c). Isopycnals along the sACCf reach their minimum depth (Fig. 4c) slightly upstream (~150°E) of the bathymetric shift near the AAR (Fig. 4d). Furthermore, convective mixing as a result of the high temperature and low salinity of fluids expelled from hydrothermal vents can enhance local vertical diffusivity and increase vertical advection, resulting in upwelling rates an order of magnitude higher than background rates within 300 m of the seafloor[34]. The AAR bloom is in a unique location where all of these upwelling mechanisms are potentially at work.

UCDW comes close enough to the surface between the sACCf and sbACC to be entrained in surface waters through deep winter mixing[35,36]. While nitrate concentrations in UCDW are uniform, DFe concentrations are highly variable. In a GEOTRACES transect to the west of our study area (along ~140°E), nitrate concentrations in the density range associated with UCDW (27.35–27.75 kg m$^{-3}$)[35,36] was 31.1–36.6 μM, while DFe

concentrations varied over a full order of magnitude from 0.07 to 0.70 nM (Supplementary Fig. 6)[37]. The variability in the DFe data shows that UCDW does not have a uniform DFe signature and indicates that point sources of Fe, such as hydrothermal vents, may influence the DFe inventory of upwelling deep water between the sACCf and sbACC.

While deep mixing during winter and upwelling throughout the spring and summer are key components facilitating the development of the AAR bloom, we contend that it is the increased Fe inventory in the upwelled water resulting from hydrothermal vent emissions that stimulates and sustains the AAR bloom each year. The Fe inventory of upwelled water determines the amount of biomass that can be supported, and we have shown that the elevated DFe inventory in surface waters at the beginning of the growing season and the entrainment of DFe-rich deep water from beneath bloom stations throughout the summer can support 2-3-fold more *P. antarctica* biomass than at stations outside of the bloom.

While the shoaling of isopycnals along the sACCf in the region of the AAR is indicative of upwelling, these isopycnals reach their minimum depth at ~150°E (Fig. 4c), far upstream of any change in NPP along the sACCf (Fig. 4a). It is only downstream of KR2 that NPP starts to increase and downstream of KR1 (the vent system much closer to the sACCf) where there is a truly substantial increase in NPP, presumably because the hydrothermal vents provide a source of Fe[7,38]. If upwelling alone, as indicated by the shoaling of isopycnals, were enough to produce a bloom, there would be increased NPP in waters where the isopycnals shoal upstream as well as downstream of the vents.

The consistent shape and position of the AAR bloom each year, visible in the NPP climatology (Fig. 1b) and annual Chl *a* images (Supplementary Fig. 4), suggest a stationary source of elevated DFe to surface waters, such as the hydrothermal vent fields KR1 and KR2. This conclusion is further supported by the climatological particle distribution from our Lagrangian satellite altimetry-based simulation. When particles are released directly above either KR1 or KR2, they closely approximate the climatological shape of the bloom. Thus, the geostrophic currents in the region indicate that the waters with elevated NPP must have passed above either KR1 or KR2, providing further evidence that these vent fields act as the source of Fe for the AAR bloom.

Depth and density profiles of DFe are also consistent with our having sampled through the nonbuoyant plume downstream of a hydrothermal vent. A plume emanating from a hydrothermal vent has two parts, the buoyant plume and the nonbuoyant plume[39,40]. The buoyant plume rises from a hydrothermal vent due to the higher buoyancy of the high-temperature hydrothermal fluids injected into the water column[39,40]. As the plume rises, the turbulence resulting from the sheer stress between the rising plume and the surrounding waters, entrains the ambient water into the rising plume until the plume is diluted and reaches neutral buoyancy with surrounding waters[39,40]. After reaching neutral buoyancy, it becomes a nonbuoyant plume, which is then advected by the prevailing currents at the depth of neutral buoyancy[39,40]. We sampled the nonbuoyant plume downstream of KR1, as evidenced by the elevated subsurface (75–300 m) DFe concentrations at bloom stations (Fig. 3a). At depths below 300 m, the difference in the concentration of DFe between bloom and non-bloom stations was much less pronounced, indicating that those depths were below the nonbuoyant plume.

We submit that further evidence of the nonbuoyant plume is visible in profiles of DFe, potential temperature, salinity, and dissolved oxygen plotted against density (Fig. 3d–g). Elevated DFe concentrations at bloom stations are associated with an anomalous water mass, between the 27.6 and 27.8 kg m$^{-3}$ isopycnals, that is colder, fresher, and more oxygenated than waters of the

same density range at non-bloom stations. These observations are consistent with the turbulent entrainment of deep waters by a buoyant hydrothermal vent plume, which will be most strongly influenced by the water mass properties of the deep waters at the depth of the vent emissions[41] (~1800–2000 m in the case of KR1), and will bear the signature of the water mass properties (temperature, salinity and oxygen in this case) at that depth[41]. The water mass properties in our depth profiles between 1800 and 2000 m, which we use as a proxy for the water mass at the same depth immediately adjacent to the vent, are colder and more oxygenated than the overlying UCDW (Supplementary Fig. 7). Therefore, a water mass generated by emissions from these vents will bear this colder and more oxygenated signal[41], which is consistent with the properties of the anomalous water mass between the 27.6 and 27.8 kg m$^{-3}$ isopycnals seen at bloom stations.

Salinity profiles are less straightforward to explain. While the salinity decreases below ~300 m (Supplementary Fig. 7b), the shift is slight, and the mixing of this water would not create a large enough change in salinity to explain the deviation observed in our density and depth profiles at blooms stations between the 27.6 and 27.8 kg m$^{-3}$ isopycnals (Fig. 3f). However, there is a clear deviation from the linear mixing line in salinity, when plotted against density, at bloom stations (Fig. 3f) between the 27.6 and 27.8 kg m$^{-3}$ isopycnals, indicative of the influence of a lower salinity water mass. In addition to carrying elevated DFe into the region of the bloom, the nonbuoyant plume may also be responsible for the stratification of the water column, creating local stratification maxima, which may be linked to the elevated vertical diffusivity[25] and therefore enhanced Fe delivery at bloom stations. While storms moving through the region would generate the same downward-propagating near-inertial waves, the vertical distribution of turbulence in the water column could be focused in the nonbuoyant plume.

Thus far, there has been only one study that observed a direct connection between hydrothermal vent activity and phytoplankton blooms[6]. The AAR bloom is different from the blooms attributed to hydrothermal vent activity downstream of the SWIR[6] in two key respects. The AAR bloom begins directly above the hydrothermal vent system as opposed to ~1200 km downstream, and the AAR bloom recurs so consistently in the same shape and location that it is visible in climatologies of Chl *a*[8–12] and NPP (Fig. 1b). We attribute these differences to multiple factors that could affect hydrothermal plume rise height, namely the vertical position of the hydrothermal vents in the water column, the position of the vents in relation to major upwelling fronts, and their arrangement relative to each other. The vents of KR1 and KR2 are substantially shallower (~1800–2000 m deep) than those associated with the SWIR blooms (3517–4170 m deep). This means that the plume does not have to rise nearly as far to reach a depth where it could be entrained into surface waters through winter mixing. The position of the AAR vents, and KR1 in particular, near the region of strong upwelling south of the sACCf means that the hydrothermal plume from this vent is being injected into a field of steeply sloping isopycnals. This suggests that the plume rise height will be greater as it travels vertically along isopycnals. It is likely that because the plume from KR1 and KR2 rises to depths so much closer to the surface, the location of the bloom is more interannually consistent. This contrasts to the blooms downstream of the SWIR where Fe is drawn to the surface by high EKE over 1000 km downstream of the vents, allowing for more deviation in the position where this Fe surfaces.

We propose that the AAR bloom is a new type of natural Fe fertilization system that receives its DFe subsidy in a way that is much different from previously described systems on the Kerguelen and Crozet Plateaus. Instead of being supported by DFe

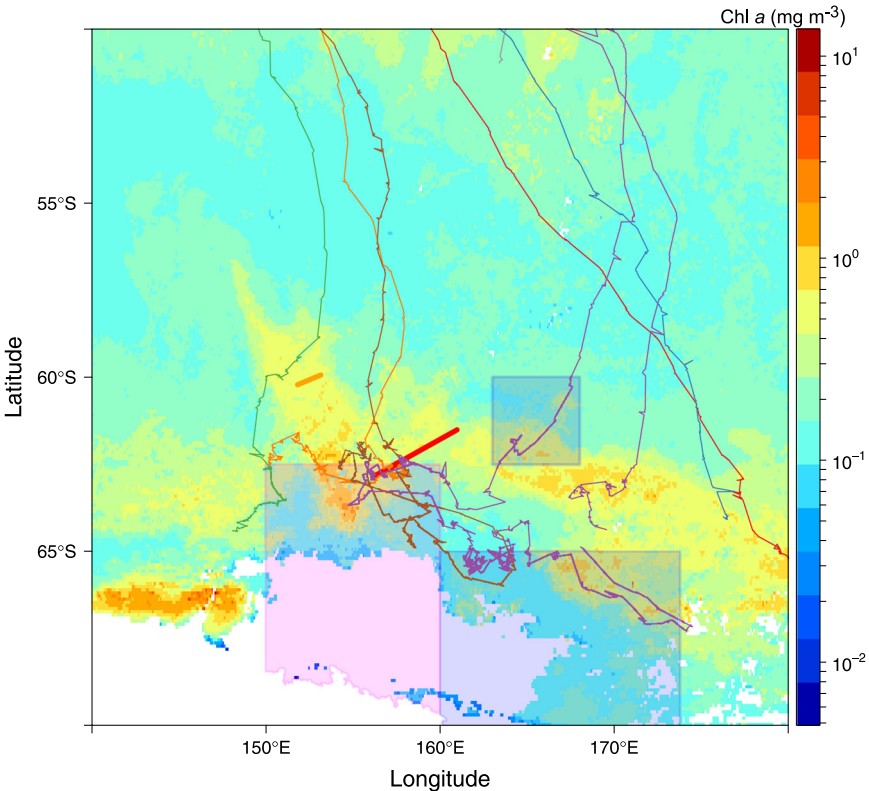

**Fig. 5 Position of Humpback whales relative to the bloom.** Map of mean chlorophyll $a$ (Chl $a$) (MODIS/Aqua) from November 2008 through February 2009 overlaid with the position of Humpback whales tagged off the coast of Eden in Southeastern Australia. Humpback whale position data are from Andrews-Goff et al.[45]. The General Protection Zone and Krill Protection Zone of the Ross Sea Marine Protected Area are indicated by the blue boxes and pink box, respectively. The white areas indicate no data due to land or persistent sea-ice and/or cloud cover, and the red and orange lines indicate the positions of the ridges KR1 and KR2, respectively[7].

from the advection of a shelf source, the AAR bloom is most likely supported by DFe from hydrothermal vent emissions. Systems receiving persistent natural Fe fertilization are marked by higher export efficiencies[20] and act as strong $CO_2$ sinks, as does the AAR bloom. Based on $pCO_2$ and wind speed measurements made during our cruise, we calculate a mean air-sea $CO_2$ flux for the AAR bloom of $-32$ mmol m$^{-2}$ d$^{-1}$ (0.38 g C m$^{-2}$ d$^{-1}$) and a maximum $CO_2$ flux of $-131$ mmol m$^{-2}$ d$^{-1}$ ($-1.57$ g C m$^{-2}$ d$^{-1}$). These values are much higher than the mean $CO_2$ flux measured at the peak of the Kerguelen bloom in November ($-17.9$ mmol m$^{-2}$ d$^{-1}$)[42] and the mean $CO_2$ flux in the Ross Sea in January ($-12.5$ mmol m$^{-2}$ d$^{-1}$)[43].

The consistent location and productivity of the AAR bloom has apparently made it a hotspot for upper trophic level activity. Tynan[44] found high concentrations of both krill and cephalopods around the sbACC, including the area encompassing the AAR bloom, and suggested that the physical processes along the sbACC provide predictably productive foraging opportunities. Humpback whales tagged near Eden, off the southeast coast of Australia, in October of 2008[45], fed primarily downstream of KR1 and KR2 in the region of elevated Chl $a$ associated with the AAR bloom (Fig. 5). Whales tagged in subsequent years along the Sunshine Coast in Australia and in Antarctica also visited the region of the AAR bloom, suggesting well-established migratory pathways and persistent use of this feeding ground[45]. The AAR bloom has implications for conservation efforts in the region. The Ross Sea Marine Protected Area covers a few small portions of the AAR bloom region (Fig. 5), however, the majority of this important foraging area remains unprotected.

The AAR bloom is the first recurring bloom that is likely to be stimulated by hydrothermal Fe to be identified in the SO. The

importance of the AAR bloom to upper trophic levels and its potential role as a $CO_2$ sink makes it essential to develop a more comprehensive understanding of this system. The SO is changing rapidly as the result of climate change, and the Pacific sector of the SO, including the region of the AAR bloom, has seen a significant decrease in NPP since 1997[12]. Interannual variability in NPP across most of the Pacific sector is linked with either changes in the number of open water days or sea surface temperature or both, but NPP in the region of the AAR bloom shows no significant relationship with either[12]. NPP in the region of the AAR bloom has decreased in response to an increasingly positive phase of the Southern Annular Mode (SAM)[12] over the last several decades[46]. However, our understanding of the changes in ocean circulation that result from a positive SAM suggest that NPP in the AAR bloom should increase due to increased upwelling of nutrient-rich deep water[47] instead of the decrease observed. Perhaps the decrease in NPP in the AAR bloom is due to light limitation of phytoplankton growth resulting from an increase in MLD[48]. Regardless, without understanding the mechanisms controlling production in the AAR bloom, we cannot predict how it will be impacted by future changes anticipated for the SO, and how those will affect the role of the AAR bloom as a $CO_2$ sink and upper trophic level hotspot.

## Methods
**Phantastic I cruise (NBP1310).** During the NBP1310 cruise aboard the RVIB Nathaniel B. Palmer, we sampled an area of the Pacific sector of the SO to the northwest of the Ross Sea, just south of the sACCf and adjacent to the sbACC from 7–15 January 2014 (Fig. 1c, d), where remotely sensed Chl $a$ concentrations were unusually high. We sampled a total of nine stations, six of which extended to a depth of 400 m and three to 2000 m. Bottom depth at our sampling stations ranged from 2762 m at station 135 to 3126 m at station 140/149 (Fig. 1c, d).

Underway measurements of temperature (Seabird SBE-38), salinity (Seabird SBE-45), oxygen (Oxygen Optode 3835), fluorescence (WET Lab AFL), and $pCO_2$ (using the Lamont Doherty Earth Observatory system[49]) were collected from the flow-through seawater system. Wind speed was measured using a Gill 1390-PK-062/R anemometer. At each station, vertical profiles of conductivity, temperature, and depth (CTD) were measured (24 Hz sampling rate) using a Sea Bird 911 plus CTD package and fluorescence was measured using a WET Labs ECO-HFL/Fl fluorometer. The depth of the upper mixed layer (MLD) was defined as depth of the maximum buoyancy frequency[50].

Discrete water samples were collected during each cast using both a trace-metal clean CTD-rosette package (TMC-CTD)[33], and a conventional CTD-rosette package. For biological sampling, water was collected at depths of 10, 25, 50, 75, and 100 m, and an additional six to seven depths between 100 and 400 m. Water samples were collected for dissolved Fe (DFe), macronutrients (nitrate, nitrite, phosphate, and silicate), Chl a, particulate organic carbon (POC), phytoplankton pigments, and simulated in situ primary production (SIS). Additionally, for the three casts conducted down to 2000 m, we collected water every 200 m below 400 m with the TMC-CTD-rosette for DFe analysis.

DFe samples were collected and filtered (Sartorius®, 0.2 μm; Satrobran 300) in a trace metal clean van. Handling of DFe samples prior to analysis was conducted on a laminar flow bench inside a positive pressure, TMC, plastic bubble. DFe samples were analyzed onboard using the automated flow injection analysis method[32,51].

Samples for macronutrients were filtered through 0.2 μm Acrodisk filters. Separate samples were taken for silicate, which were stored at 4 °C, and for nitrate and nitrite, which were stored frozen. After the cruise, nutrient samples were analyzed colorimetrically on a Bran and Luebbe trAAcs 800 Autoanalyzer[52]. Measurements were made for silicate, nitrate and nitrite together, and nitrite separately. All measurements were calibrated with standards diluted in low nutrient seawater, which was also used as rinse water between the samples.

Samples for POC analysis were filtered onto precombusted (450 °C for 4 h) 25 mm Whatman GF/F filters. Filters were dried at 60 °C for analysis on a Costech Elemental Analyzer using acetanilide as a calibration standard[53].

Samples for Chl a were collected and analyzed on board with a Turner Model 10 AU fluorometer using the acidification method[54]. Depth-integrated Chl a was calculated down to 100 m at all stations by linearly interpolating between discrete Chl a samples. An integration depth of 100 m was chosen to ensure that all phytoplankton biomass was accounted for and to enable direct comparison with depth-integrated Chl a values reported elsewhere in the SO. Below 50 m, Chl a concentrations were very low and contributed a negligible amount to depth-integrated values.

Samples for HPLC analysis of phytoplankton pigments were filtered onto 25 mm Whatman GF/F filters, flash-frozen in liquid nitrogen, and stored at −80 °C until analysis within six months of collection. Filters for pigment analysis were extracted in 90% acetone (48 h at 4 °C) and then separated by HPLC (Waters 2695, with a Zorbax Eclipse XDB-C8 column, 3.5 μm particle size)[53]. Detection was based on retention time and diode array spectroscopy (Waters 996) at 436 nm[53]. The CHEMTAX analysis package (version 1.95)[55,56] was used to assess phytoplankton class abundance[57].

Depth-integrated daily NPP was assessed at four different stations (119, 130, 140, and 150) using [14]C-based SIS production on-deck incubations[58]. Depth-integrated NPP was calculated to the depth of the 1% light level. The depth of light levels used for the integration (85%, 65%, 25%, 10%, 5%, 1%) were calculated using Beer's Law and a diffuse attenuation coefficient that was a function of the mean Chl a concentration in the mixed layer at each station[59]. The depth of the 1% light level at stations 119 (87 m) and 140 (107 m) outside of the bloom was much deeper than at bloom stations 130 (30 m) and 150 (35 m).

The vertical flux of DFe across the ferricline for each station was calculated by multiplying the DFe concentration gradient by the mean vertical diffusivity. Vertical diffusivity ($K_z$) was calculated from vertical density profiles using Thorpe-scale analysis[33,60] (additional details provided in the Supplementary Material). The DFe concentration gradient for each station was calculated as the slope of the ferricline (defined as the difference between the minimum DFe concentration between 15 and 56 m and the maximum DFe concentration between 150 and 205 m) divided by the thickness of the ferricline (defined as the vertical interval between the minimum and maximum DFe concentrations). Mean vertical diffusivity was calculated as the mean $K_z$ in the ferricline. Station 135 was excluded from this analysis because it had an anomalously high vertical diffusivity coefficient that was more than two standard deviations above the mean.

Air-sea $CO_2$ flux inside the bloom was calculated for each $pCO_2$ measurement where underway Chl a fluorescence exceeded 5 mg m$^{-3}$, using the corresponding shipboard measurements of wind speed and underway temperature, salinity, and the partial pressure of $CO_2$ in water ($pCO_{2w}$) and the atmosphere ($pCO_{2a} = 400$ μatm)[61]. Air-sea $CO_2$ flux (negative values denote flux into the ocean) was calculated as[62]:

$$F = kK_0(pCO_{2w} - pCO_{2a}) \tag{1}$$

where $k$ is the gas transfer velocity and $K_0$ is the solubility of $CO_2$ in seawater[63].

**Satellite imagery.** MODIS/Aqua Chl a images were obtained daily during the cruise and were used to target areas of high Chl a. To obtain bloom information

after the cruise, we used monthly Chl a concentrations from level 3 MODIS/Aqua data (4 km resolution; Supplementary Fig. 1). Additionally, monthly Chl a (Oct–Mar) from SeaWiFS (1997–2002) and MODIS/Aqua (2002–2019) were used to calculate images of mean seasonal Chl a (Supplementary Fig. 4). Monthly SSMIS (Special Sensor Microwave Imager/Sounder) sea ice concentration data, taken from the NOAA/NSIDC Climate Data Record of Passive Microwave Sea Ice Concentration, Version 3[64], were used to track the receding sea ice edge, defined as the 50% sea ice concentration contour (25 km resolution). Satellite-based NPP was calculated using the algorithm of Arrigo et al.[11] (details in Arrigo et al.[29]) using IDL (Interactive Data Language, L3Harris Geospatial) and SeaDAS version 7.5.3 (NASA). Geostrophic currents derived from satellite altimetry were produced by Ssalto/Duacs and distributed by the European Copernicus Marine Environment Monitoring Services. This multi-satellite global product provides daily velocities with a 1/4° spatial resolution.

**Lagrangian plume modeling.** A lateral advection scheme based on geostrophic currents from satellite altimetry was used to model the Lagrangian plume downstream from the hydrothermal vents. The plume was estimated by reproducing the dispersion pathways (using satellite-based geostrophic currents) during the high NPP period (Nov–Apr)[65]. Lagrangian trajectories were derived by applying a Runge–Kutta fourth-order scheme with a time step of six hours to the velocity fields, which are linearly interpolated in both space and time. For each advection period, the model identified the most recent contact of each water parcel to the area at the surface directly above the hydrothermal vents. As in Sergi et al.[65], hydrothermal vents were identified with disks of 50 km radii centered in the hydrothermal vents' locations. For each phytoplankton growing season from 1997–2019, a map was produced every 15 days between 1 November and 30 April. The climatological signal (Supplementary Fig. 3) was created by taking an average of each annual map.

**Argo-derived data products.** The Argo New Displacements Rannou and Ollitrault (ANDRO) dataset uses the displacement of Argo floats at their parking depth to map global ocean circulation[66]. We used the 3° × 3° resolution gridded climatology data to look at eddy kinetic energy (EKE; calculated as one half of the sum of the standard deviation of the meridional velocity and the zonal velocity) in our region of interest.

The Monthly Isopycnal/Mixed-layer Ocean Climatology (MIMOC) provides global monthly ocean property maps at 0.5° resolution[67]. The maps are based primarily on Argo CTD data, supplemented by shipboard and ice-tethered profiler CTD data. The optimal interpolated (0–1950 dbar) sigma-gridded data for January were used to plot the zonal change in the pressure associated with different isopycnals.

**Statistical analysis.** A two-sample t-test was used to compare DFe and $K_z$ values at stations inside the bloom to those outside the bloom for different depth bins. A Shapiro–Wilk normality test was used to evaluate the normality of the variable distributions. Linear regression analysis was used to evaluate the relationship between vertical DFe flux, vertical DFe gradient, and depth-integrated Chl a. Statistical analyses were considered significant for p values < 0.05. Statistical analysis was done using R version 3.6.2 (R Project for Statistical Computing).

**Reporting summary.** Further information on research design is available in the Nature Research Reporting Summary linked to this article.

## Data availability

The satellite data that support these findings are freely available and may be downloaded from the links provided here: MODIS/Aqua Chl a images are available from https://oceancolor.gsfc.nasa.gov/, SSMIS sea ice concentration data are available from https://www.nsidc.org/data/g02202, SSALTO/DUACS global gridded sea surface height (product id SEALEVEL_GLO_PHY_L4_REP_OBSERVATIONS_008_047) is available from https://marine.copernicus.eu, the ANDRO dataset is available from http://www.coriolis.eu/Data-Products/Products/ANDRO, and the MIMOC dataset is available from https://www.pmel.noaa.gov/mimoc/. The cruise data that support these findings are available at the Stanford Digital Depository (permanent URL: https://purl.stanford.edu/sn954dk6470).

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

## Acknowledgements

We thank the captain and crew of the RVIB Nathaniel B. Palmer for their assistance during the cruise. We also thank all the members of the Phantastic I research team, whose tireless work contributed greatly to this research. This work was supported by the National Science Foundation, Office of Polar Programs (ANT-1063592).

## Author contributions

C.M.S.S., A.-C.A. and G.v.D. conducted the data analysis. C.M.S.S. wrote the paper. K.R. A. wrote the original grant proposal. L.J.A.G. and P.L. collected and analyzed trace metal samples. H.v.H. calculated vertical diffusivity. W.H.v.d.P. analyzed pigment samples. S.S. provided the Lagrangian particle tracking results. All authors contributed to the ideas and commented on the manuscript.

## Competing interests

The authors declare no competing interests.
