## [Peer Review File · Nature Communications]

REVIEWER COMMENTS

Reviewer #1 (Remarks to the Author):

The manuscript presents remote sensing and in situ dissolved iron and biological measurements to characterize a massive phytoplankton bloom that occurs annual north-west of the Ross Sea, south of the Macquire Ridge system. The manuscript presents new insight into the processes that lead to the development of this large phytoplankton bloom. I guess my one contention with the manuscript is the idea that the main source of iron fueling the bloom is from hydrothermalism—certainly bottom water torque which can enhance water exchange between deep and shallow water masses (Rintoul, 2018). The authors Kz data strongly supports that idea. But the apportioning of the iron source is, in my view, a bit speculative. The reason I say this is that the hydrothermal system located on KR1 is situated at a depth between 1.7-1.9 km. The concentration of iron and manganese released from these plumes is up to 150 and 100 nM, respectively (Hahm et al., 2015). In the present study, elevated iron and manganese concentration were not measured at depth. The concentrations for these two elements are background levels of 0.4 and 0.2 nM, respectively. Indeed, I would have expected elevated manganese as depth and not just within or just below the euphotic zone. Middag et al. (2011) – fig 2 – found elevated manganese throughout the water column for the Triple Junction Ridge Region hydrothermal system near Bouvet Island, in the south Atlantic section of the Southern Ocean. Away from the hydrothermal system, they found a peak in manganese concentration below the euphotic zone, similar to what is seen in the present study. Could the elevated manganese concentration in the surface ocean be a result of photochemical reduction of particulate manganese? (Sunda et al., 1983).

In light of my comments, I would suggest that authors explore other possible sources for iron at the bloom site and consider the following – 1) perhaps the sampling regime missed the elevated iron and manganese vent sites?; 2) Is there any chance that the eddies that form in the region as a result of water being squeezed through and south of the Macquarie ridge could be a source of iron? Eddy activity in the region can be high – the bloom site is close to the southern ACC front (sACCF) where eddy activity is expected to be high (Frenger et al., 2015; Rintoul, 2018). Perhaps have a look at the following references - (Rintoul et al., 2014; Schuur et al., 1998)

It may well be that hydrothermal venting is the source of the iron, but I guess I'd like this to be firm up. Regardless, I think this manuscript is well written and provides valuable insight into bloom formations in the Southern Ocean.

References

Frenger, I., Münnich, M., Gruber, N., Knutti, R., 2015. Southern Ocean eddy phenomenology. *Journal of Geophysical Research: Oceans*, 120(11): 7413-7449.

Hahm, D. et al., 2015. First hydrothermal discoveries on the Australian-Antarctic Ridge: Discharge sites, plume chemistry, and vent organisms. *Geochemistry, Geophysics, Geosystems*, 16(9): 3061-3075.

Middag, R., de Baar, H.J.W., Laan, P., Cai, P.H., van Ooijen, J.C., 2011. Dissolved manganese in the Atlantic sector of the Southern Ocean. *Deep Sea Research Part II: Topical Studies in Oceanography*, 58(25): 2661-2677.

Rintoul, S.R., 2018. The global influence of localized dynamics in the Southern Ocean. *Nature*, 558(7709): 209-218.

Rintoul, S.R. et al., 2014. Antarctic Circumpolar Current transport and barotropic transition at Macquarie Ridge. *Geophysical Research Letters*, 41(20): 7254-7261.

Schuur, C.L. et al., 1998. Sedimentary regimes at the Macquarie Ridge Complex: Interaction of Southern Ocean circulation and plate boundary bathymetry. *Paleoceanography*, 13(6): 646-670.

Sunda, W.G., Huntsman, S.A., Harvey, G.R., 1983. Photoreduction of manganese oxides in seawater and its geochemical and biological implications. *Nature*, 301(5897): 234-236.

Reviewer #2 (Remarks to the Author):

This is very well written paper looking at the iron biogeochemistry underpinning a persistent phytoplankton bloom in the Southern Ocean. The study focusses on a persistent bloom in the Pacific sector of the Southern Ocean and demonstrates that the bloom results from a combination of factors consisting of elevated subsurface iron concentrations, increased diffusivity and a bloom dominated by an efficient user of iron. The consistent occurrence of the bloom then supports further productivity at higher trophic levels and makes it an important feature in the Southern Ocean.

The authors link the elevated subsurface Fe to hydrothermal inputs from the Australian Antarctic Ridge. The novelty of the work is in the identification of further persistent blooms supported via hydrothermal inputs, and a consequent strengthening of the link between hydrothermal inputs of iron and surface productivity. This work is not the first to identify hydrothermal sources of iron as potentially important for productivity in the Southern Ocean. However, the importance of hydrothermal sources of iron to surface productivity is still a little controversial. This study is the first to use in-situ dissolved trace metal data to explore the link.

I want to emphasize here that I agree that the data support the role of increased iron fluxes from below the ferricline in fuelling the enhanced productivity in the region. The higher diffusivity, low geostrophic currents and potential upwelling combined with the extremely efficient iron use efficiency for *Pheaeocystis antarctica* thus seem to be very important for maintenance of the bloom and further suggest that an interesting interaction between physical and biological factors is at play. In my opinion this is already a significant and important observation.

The authors go on to use depth profiles of dissolved Mn and Fe to support the link to a hydrothermal source, ruling out ice (although the bloom starts close to the ice according to figure S1) and other sources in the process. Unfortunately, I am not convinced that the Mn data fully supports the hydrothermal source hypothesis. Indeed, Mn concentrations are enhanced below 90 m, but this is a relatively shallow maximum that is also observed in other regions of the Southern Ocean (e.g. the GIPY05 Mn transect shows that both hydrothermal influenced Mn profiles and non hydrothermal influenced Mn profiles produce the same kind of subsurface maxima). So I am struggling to really link this maximum to a hydrothermal source just 168 km away at >2000 m depth. There is clearly still a maximum in the Mn profile outside the bloom area - albeit not so pronounced (but at the same depth which would be strange if this feature was really linked to hydrothermal sources?). Furthermore there is no difference in Mn concentrations at depth. To me the Mn profiles are suggestive of an enhanced remineralisation/scavenging mechanism likely driven by the higher productivity. So overall I do not think Mn is supporting the hypothesis, and potentially even pointing to a different mechanism for sustaining iron in this region. Furthermore station 133 is presented as grey and thus outside the bloom, but it looks like it is from inside the bloom on the map given in figure 1d.

Given my concerns raised above, although I do find the study technically sound and important for the field, I need more evidence with respect to the requirement for a hydrothermal source for explaining the persistence of this bloom.

Specific comments:

Line 62: How deep are the vents in the region and close to the bottom were the 2000 m casts?

Line 153: Figure 1 panels are referred to out of order and Fig 1a not referred to at all as far as I could tell.

Line 178-182: nutrients not completely drawn down – so iron still limiting?

Line 194: What was the mixed layer depth?

Line 202/figure 3 caption: vertical diffusivity/K – use vertical diffusivity in caption of fig 3 too.

Line 218: So actually the difference here is linked to the species growing in the bloom and their high iron use efficiency? Why do the authors think it is Phaeocystis and not diatoms growing in the bloom?

Line 261: The message I get from figure S1 is that the bloom forms close to the sea ice before the ice retreats, leaving the bloom stranded. Would remineralisation have to be unreasonably efficient in order to maintain the DFe gradient?

Line 274-281. The enhanced diffusivity is key to maintaining the supply of Fe to the surface. Reasons behind the high diffusivity are not mentioned here, but I think they should be.

Line 284. The potential possibly? Remove one or the other.

Line 307. Productivity enhanced by upwelling is commonly associated with enhanced remineralisation. Maybe the Southern Ocean is different, but was this scenario considered as a potential mechanism for resupplying remineralised Fe to the surface?

Line 347-349. But Mn is not elevated in deeper waters compared to surrounding waters (FigS2).

Reviewer #3 (Remarks to the Author):

Review of Schine et al

This is a very nice paper documenting a recurring area with a large phytoplankton bloom near the southern boundary of the Antarctic circumpolar current. This is an important area for carbon export from the global ocean where productivity is often regulated by Fe availability. It is also an important feeding area for marine animals. These authors suggest that this Fe is provided by hydrothermal activity on the nearby Australian-Antarctic Ridge. This is a very important finding. The provision of Fe by hydrothermal sources is an open question in the Southern Ocean and this paper does a very nice job of documenting how this might look and what the impacts might be. I find this to be a fascinating and important paper. Hydrothermal Fe is a persistent source over time, while ice cover and other sources may have long and short term variability. Even then, the question of how the Fe gets to the surface ocean from hydrothermal vents is an important one. This transport may be impacted by deep water formation due to changes in climate. This is a very important contribution to the scientific literature.

While I like the manuscript as a whole, I have many issues with the work as presented, but they are all addressable. So, while the bulk of my review may come across as negative (and long), please do not allow those comments to obscure my enthusiasm for the ultimate publication of this manuscript. I strongly recommend publication of this manuscript but only after significant modifications are made.

The following are places where comments about data interpretation that I find most troubling:

1. The calculations of K_z and thus vertical Fe flux are very important parts of this manuscript. In these calculations you make many assumptions. In the end, you calculate that that it is much larger than elsewhere in the southern ocean. I address the use of K_z below in my extended comments below. Those comments must be addressed. The error and error bars must be addressed and added to figure 3c.
2. You must define depth integrated. Over what depth interval? This is very important when you discuss ratios of Fe to other things. If you integrate the Fe over the full water column, you would get one answer and if you did it over the upper 100 m you would get another answer. For POC and chlorophyll, are you integrating from 0 to 50m or to 100m? If you integrated both POC and chlorophyll and Fe to 50 m then the ratios would be one thing and if you did all of them to 100 or 150m those ratios would be very different. This also plays into your indiscriminate use of the concept of “mixed layer depth”. What was the mixed layer depth during this study? Is it 40 m or 100m or 150m? You suggest late in the manuscript that the deepening of the MLD results in decreased productivity. I assume that you are thinking of below 100M? So, what is your mixed layer and how does it relate to all of the above?
3. I believe that there is a significant problem with figure 3c and the associated statistics. The data density between the different profiles is significant especially for stations 120 and 136. I can imagine that if the authors made use of error bars that the error for those two stations should be rather large. The authors did not provide their data (the link did not work), so I am left to do evaluations by eye. In my imagination, most of the POC and Chlorophyll are in the upper 40 m, which means that integrated values at 120 and 136 are each defined by a single data point. May be there is a lot of POC and Chlorophyll at depth. Stations 120 and 136 do not appear in figure 3a and b, but do in c. In the end, I don't think that these data support the use of a regression line on an x-y plot. This plot, if retained must have x and y error bars. I address this further below in more detail.
4. The manuscript makes comparison of various rates and processes in the AAR plume versus elsewhere in the Southern Ocean. Are you comparing the biggest values in each of the systems

(AAR vs Crozet and Kergeulen) or are you cherry picking the “off-shelf” values. How are you selecting off-shelf versus on-shelf? Are you always picking the off-shelf values? I’m not sure how you want to parse this, but by the end of the paper it is very unclear what you are comparing to what. I think that these elevated values out in the middle of ocean (AAR) are pretty important, however how they compare to other blooms is important as well. But how those comparisons are being made need to be clear.

5. There are many very general terms in this manuscript that need to be more scientific. The words “almost” and “most” should be quantified. The manuscript makes extensive use of the following undefined terms “off-shelf,” “depth integrated,” and “mixed layer depth (MLD).” Each of these concepts are central to the manuscript and must be defined, as in what is the depth range for integration, what is the actual MLD?
6. Data must be made available.
7. Figures, either text or supplement need to do a better job of showing depth resolved Fe and Mn.

The comments below address the ideas discussed above in more detail and also addresses other areas of important consideration.

L112 this is a very important section in the paper because much of the paper depends on this calculation. You need to tell us how K_z was calculated other than a method name and a reference. What data did you use; what are the most important parameters/variables in this calculation? In figure 3B the K_z values look discrete but the CTD data were collected at 24 Hz. Why is this? Are the data binned? From what depth did you choose your K_z for the vertical Fe flux? L118 what is the “same” depth range? You are providing almost no information to your reader and your results are not reproducible without more information here or in the supplement. If your readers are to understand what the divers are for the large K_z then you need to tell them what those variables are and how they are used. My examination suggests a rather large error for K_z and thus for vertical Fe flux. You make a significant investment in this number and you tell us that it is much larger than elsewhere in the southern ocean. Why is this? Help us understand. Maybe the primary equations from reference 26 can be shown in the supplement?

For Fe gradient why did you pick the depths that you picked? Elsewhere in this review I ask about your MLDs. That information might help us understand some of your rationales for K_z and vertical Fe flux. I have trouble appreciating how the lowest Fe value in the upper 50 meters has any relationship to the highest values between 150 and 200m. I think that the MLD plays a role here. You must show x and Y error bars in figure 3c.

L158 what were the speeds of those currents? Do we have a sense of the residence time of the water in the bloom area? If it is three months, then perhaps you need to work on your explanations, if it is 3 weeks, then this strongly supports your suppositions and conclusions. Has the bloom also been transported in this direction? I guess that we can get a sense of this from figure S1? This is important from my perspective because you are not immediately above the ridge crest and your deep Mn does not support your shallow (see comment below).

Figure S1 It would be pretty nice to see this for another year of ice break-up to demonstrate your point that ice is not the Fe source.

L161-175 Great section. Lots of very strong data. However you need to clarify what you mean by inside and outside the bloom area. In general you are consistent but line 167 suddenly discusses the edge of the

bloom versus inside and outside. In fact, it is not until L 174 that we get an idea that 119 and 140 are outside. Maybe this needs to be stated at the start of this paragraph?

L168 and 171 “depth integrated” Please define the depth range for depth integration. You may need to address the concept that most of the PC and Chlorophyll are shallower than 50 m if your integration goes deeper in water column

L172 and 173 It seems that biomass is an arbitrary unit in this case? IS it solely from the satellite data? Are units needed on biomass, did you measure it? Would chlorophyll be a better word? Forgive me, I’m no biologist, but the introduction of a seemingly unquantified variable in a well quantified section seems odd to me.

L184 any insight why *Phaeocystis antarctica* is dominant here?

L198-201 Please present Mn data in the same way that you present the Fe data so that we can see the Mn in the shallow ocean. Please provide insight on why Mn is depleted in the surface ocean. It would seem that once the Fe was taken up that there would be a relic Mn signal. Where is that Mn? Does this point to Mn limitation or any such thing? Unfortunately the link to the data did not work, so I wasn’t able to appreciate the Mn data in all of its glory. Also, while upwelling Fe might support the elevated subsurface Fe levels, this does not work for Mn. You must discuss why this is. Is it because the surface water has transported the upwelled Fe and Mn away from ridge crest and that you are not sitting directly over the zone of upwelling Fe and Mn? Is it a water mass question?

L202-204 I am probably missing something here. All of your productivity is in the upper 40m, so do we need to discuss how we get the Fe into that depth range? Or is it sufficient to bring it up to 90 m? What is the mixed layer depth? Nitrate, Chlorophyll and POC all suggest 50m. Please provide +/- on vertical diffusivity.

Also, why is this true? Why is vertical diffusivity inside the bloom higher than outside? This is not inherently obvious. I understand that the Fe gradient would drive a higher Fe flux but the increase in vertical diffusivity needs to be explained.

L206-219 Again, I am obviously missing something. All of the productivity is in the upper 40 meters but you are calculating Fe uptake down to 150m. This seems like a huge problem. When I integrate chlorophyll or POC, then the numbers won’t change beyond 40 m??? However you integrate the missing Fe over 150 m. This really changes all of your calculations. How is this biomass removing Fe from deeper in the ocean? You really need to help your reader understand why this approach is valid.

L225 this is somewhat less than the number quoted on L216 for diatoms. Somehow these two sets of ratios (inside and outside) are much closer than for POC than they were in section above. IS this important? What does it reflect? Overall, I have trouble appreciating the importance of this paragraph.

L230-232 Vertical Fe flux is a function of both vertical diffusivity and the Fe-gradient. So why doesn’t integrated chlorophyll correlate with all three variables? Something is strange here. Again, why is this important, and what does it mean? Where are the data from station 136; they don’t show up in figure 3 a and b but do in c? Same is true for stations 133 and 120. Stn 133 doesn’t even show up in Figure 2. In Figure 3c, 136 (and 120) appears to be an outlier? Fe gradient looks smaller for 136 but larger for 120, what about the diffusivity? Integrated chlorophyll is based on a single data point for 136 and 120? Hard to tell what the numbers are because your data set is not available and ODV is well known to create data like it has for the shallow chlorophyll at these stations. Is it reasonable to estimate the integrated chlorophyll without the shallower data points? Again, I don’t have the data set but ODV suggests that the deeper ones are largely irrelevant? So, again, I ask how did you do this integration?

Does station 136 belong in your figure 3c statistical analysis. I can guarantee you that if you remove this data point that your correlation would improve. The same point should be made about station 120 and 133. In the end, the statistical validity of this entire section seems to be dubious at best. You really need to rethink this. What you are saying may be true. The increase in chlorophyll (and POC?) with increased Fe flux seems reasonable, but your data may or may not support the use of stations 120, 133, and 136 in this analysis. I suspect a table would be more helpful than this graph. You have two overlapping data clusters, not a trend.

L242, 247 Doesn't most of the Fe from Kerguelen ultimately come from the shelf? Are you talking about proximity to the shelves? Are you dismissing the shelf-blooms? Are you instead wanting to say that your blooms are the largest in the open ocean away from shallow topography and margin/shelf influence? I can see this importance, but it seems odd putting it in terms of a negative feature (non-shelf). I suggest that you refer to this in a more positive sense. Shelves are important. I understand what that you want to say that this bloom is not near any shelf. I would think about how to present this. It's not a deal breaker, but it was hard to figure out what you were wanting to say. To this reader it was very distracting to understand what your backwards logic was trying to express. L249 How do your outside the bloom values compare to the rest of the SO? L250 What part of Crozet? What depth? Are there shelf and non-shelf values at Crozet that you are comparing to? This whole section really makes it difficult to know what is being compared. At this point I think that you might need a table!

L272, you MUST say why your Kzs are higher than elsewhere. What is the physical reason? IS it knowable? This is a great reason to show this calculation somewhere so that we can have faith that this order of magnitude is real. Gut instinct of your reader is that you made a mistake. Because this is a major major part of your story, you must make it so that the reader can understand it.

The Case for hydrothermal DFe (L 328)

You might want to consider Tagliabue and Resing 2016 in this set of arguments. They seem to show that the isopycnals crossing this section of the ridge crest are similar to those that are also upwelling in your study area. They also make arguments about the importance of hydrothermalism in the southern ocean versus elsewhere. While your deeper Fe supports your shallower Fe, the same is not the case for Mn. Is this because you are not in the "stem" of the up welling? Perhaps Fe and Mn are upwelling upstream and the surface waters move in a different direction at a different speed than in the deeper waters? Can you address the dMn minimum in the surface ocean? Can the hydrothermal Fe and Mn accumulate under the ice during the winter? You need to make a case for there being more Mn in the near sub surface than at depth. I don't think that it can be light driven recycling. Mn is typically low in the southern ocean and deep mixing will ultimately mix out the Mn on a seasonal basis. Perhaps I am incorrectly assuming that the residence time of Mn in this region is short compared to the rest of the ocean.

Tagliabue A, Resing J. 2016 Impact of hydrothermalism on the ocean iron cycle. Phil. Trans. R. Soc. A 374: 20150291. <http://dx.doi.org/10.1098/rsta.2015.0291>

L383-386 Here is the whole shelf/ non-shelf set of questions again? Are you only comparing AAR to the lower off-shelf values at Crozet and Kerguelen? I find my self to be very very confused about this issue.

Do you have a sense of the total flux (rate x area x time length) or total flux per m² (rate x time length) for the different systems?

L 387-396 I like this section. I thought it was gratuitous when I first read it. However upon contemplation I realize that climate change could weaken both down and upwelling, decreasing Fe input and the amount of forage available.

I guess I would like to see the conclusions address the hydrothermal input and the unique situation of the front over the ridge crest. I would like it to address the concept that hydrothermal sources are persistent

while other sources may experience seasonal, annual, decadal, and longer variability. On the one hand the persistent source is important, on the other, if it can't get to the surface ocean, then it is less so.

Details to be addressed:

Abstract

The word unprecedented means "never done or known before." You proceed in the paper to document several regions in the southern ocean where similar observations of recurring blooms in specific geographic areas. You might want to rethink the use of this word. You don't need this to convince me that your observations are important.

L13 later in the paper it is "-32" The units should be in $\text{mg d}^{-1}\text{m}^{-2}$, I hope.

L33 Ardyna et al used surface drifters in conjunction with historical 3-He data. 3-He was not measured as a part of their study. Your sentence needs to better reflect this.

L37 Do you accept their hypothesis of enhanced eddy kinetic energy? Should that apply here? Why don't you calculate it?

L47 If the records exist, and you do reference them, can you need to quantify "almost". Almost is not a great word for a scientific paper.

L82 what do you mean when you say "Samples were digested for 4 h....." please provide a reference or a description.

L84 Did you distill reagent grade acid? How did you do it? Was it a sub-boiling distillation or a 2x quartz distillation? Perhaps you purchased the acid.

L86 do you mean a serial dilution of a 1000ppm standard? It seems very difficult to use a 1000ppm standard to spike your samples. If it was a purchased standard, can you provide the manufacturer?

L134 please define SSM/I

L138-139 It would seem that a single Arrigo reference would suffice.

L 254 to 257 Is this time scale information needed for anything other than information? It seems all three time length scales are the same, so what can the reader do with this information? Maybe it belongs in the introduction?

L262 Really need to define the mixed layer. It keeps showing up, but with no definition.

L291 please use "Advection from continental shelves...."

L296 While I do agree with your assessment, I note that the ocean margins and associated slopes do go to 2000m.

L334 do you have an actual +/- value for "same location" Perhaps not important???

L340 the referenced paper does not report dissolved Fe. It reports unfiltered total HCl leachable Fe and Mn.

L372 Good point, again I would consider Tagliabue and Resing reference from above.

L378 I would use "from the advection of a shelf source,"

Figures

Figure 1. IS the white in the figure ice or land? Could you please add that to the caption?

Figure 2 Where are data from station 133.

Supplement should show all vertical profiles. ODV makes up data and puts features where there are none. This is happening in panel a and b, but I can tell that the gridding is more appropriate in panels c and d.

The greater data density in c and d (vis a vid stations 120 and 136) allow you to more honestly grid your data. In addition, you integrate the water column, without telling us what that means. How deep did you go? Do Chlorophyll and POC go to zero? Is the upper water column extremely more important than below 40m? I can't tell. SO the addition of vertical profiles at the appropriate depth range and a definition to the depth range of integration are crucial. If there is significant chlorophyll and POC at depth, then you might consider using a stretched color scale in ODV.

Figure 3 a. Where are the data for stations 120, 133, and 136?

b. putting a line through these data is a bit of a stretch. Better to make two groups and take averages and standard deviations. While your p-value may tell you one thing, looking at the data tells me that the relationship is not strong enough for adding in a line.

Figure 4. IS the white in the figure ice or land? Could you please add that to the caption?

Figure S2 Do you have all of the data?

Figure S3 If these data have not been published, have you been in contact with the authors about using them in this fashion? Maybe you need to reference the International Geotraces data product?

Can you put in a comparable ODV plot of Mn in the upper 400 m in the supplement?

This review was completed by Joseph Resing

We are enormously grateful for the comments on our manuscript from reviewers. We feel that this new version of the paper is much stronger as the result of the comments we received on the original manuscript. We have addressed all of the comments and have detailed our response to specific comments below. Our response to each comment is bulleted and in italics below the relevant comment. Overall, we have added several new pieces of evidence to strengthen the argument that the iron feeding this bloom has a hydrothermal origin. This involved reworking the “Case for hydrothermal DFe” section of the Discussion. We have added new field data showing the differences in the water mass where we see elevated iron as well as information on isopycnal depth and eddy kinetic energy from datasets derived from Argo float data. We have added additional panels to Figure 3, and Figure 4 is entirely new. We have also added Supplemental Figures S2, S3, S4, and S6. We provide more details about this added information in our responses to specific comments. All of the changes in the manuscript and in the figure captions have been highlighted in yellow.

Our study raises several very interesting questions about phytoplankton community composition and elevated turbulence in the region of the AAR bloom. We have added a theory on the origin of the elevated K_z, and we now mention that Phaeocystis may be the dominant phytoplankton in this bloom due to their lower iron requirement. These are very interesting questions that require further investigation, but we do not have any additional data to fully answer them, and lengthy speculation is not possible.

We have opted to remove the DMn data from the manuscript. These data raised many questions from reviewers that we were unable to answer. We feel that the support that the DMn data provided for the argument that the DFe feeding the bloom was of hydrothermal argument was weak and has been replaced by much stronger arguments detailed in the “Case for hydrothermal DFe” section of the Discussion.

REVIEWER COMMENTS

Reviewer #1 (Remarks to the Author):

The manuscript presents remote sensing and in situ dissolved iron and biological measurements to characterize a massive phytoplankton bloom that occurs annual north-west of the Ross Sea, south of the Macquire Ridge system. The manuscript presents new insight into the processes that lead to the development of this large phytoplankton bloom. I guess my one contention with the manuscript is the idea that the main source of iron fueling the bloom is from hydrothermalism—certainly bottom water torque which can enhance water exchange between deep and shallow water masses (Rintoul, 2018). The authors K_z data strongly supports that idea.

But the apportioning of the iron source is, in my view, a bit speculative. The reason I say this is that the hydrothermal system located on KR1 is situated at a depth between 1.7-1.9 km. The concentration of iron and manganese released from these plumes is up to 150 and 100 nM, respectively (Hahm et al., 2015). In the present study, elevated iron and manganese concentration were not measured at depth. The concentrations for these two elements are background levels of 0.4 and 0.2 nM, respectively. Indeed, I would have expected elevated manganese as depth and not just within or just below the euphotic zone. Middag et al. (2011) – fig 2 – found elevated manganese throughout the water column for the Triple Junction Ridge Region hydrothermal system near Bouvet Island, in the south Atlantic section of the Southern Ocean. Away from the hydrothermal system, they found a peak in manganese concentration below the euphotic zone, similar to what is seen in the present study. Could the elevated manganese concentration in the surface ocean be a result of photochemical reduction of particulate manganese? (Sunda et al., 1983).

- We have removed the DMn data from the manuscript but have added other evidence suggesting that the DFe supporting the bloom is hydrothermal.*

In light of my comments, I would suggest that authors explore other possible sources for iron at the bloom site and consider the following –

- 1) perhaps the sampling regime missed the elevated iron and manganese vent sites?;
 - We have added details on hydrothermal plume dynamics and describe our sampling regime relative to the positions of the buoyant and nonbuoyant plumes and the evidence for this in our depth and*

density (new figures) profiles of DFe, as well as Temperature, Salinity and Oxygen. This is all in the “Case for hydrothermal iron” section in the Discussion.

- 2) Is there any chance that the eddies that form in the region as a result of water being squeezed through and south of the Macquarie ridge could be a source of iron? Eddy activity in the region can be high – the bloom site is close to the southern ACC front (sACCf) where eddy activity is expected to be high (Frenger et al., 2015; Rintoul, 2018). Perhaps have a look at the following references – (Rintoul et al., 2014; Schuur et al., 1998)
- *We have added two figures (Figure 4 and Supplemental Figure S3) that clearly show that the region of the AAR bloom is marked by very low EKE, and therefore the action of eddies is not influencing the formation of this bloom.*
 - *However, we have also added new information about isopycnal depth and look at the shift in the depth of isopycnals along the sACCf (also Figure 4 and Supplemental Figure S3). We clearly see a shallowing of isopycnal depth associated with the AAR, which would indicate an influence of bottom pressure torque as the ACC flows over the AAR. We discuss this in the “Case for hydrothermal DFe” section of the Discussion.*

It may well be that hydrothermal venting is the source of the iron, but I guess I'd like this to be firmed up. Regardless, I think this manuscript is well written and provides valuable insight into bloom formations in the Southern Ocean.

- *We have reworked the entire “Case for hydrothermal DFe” section with several new pieces of information. We include details of isopycnal depth, an anomalous water mass indicative of the nonbuoyant plume from a hydrothermal vent associated with elevated DFe concentrations, and the results from a Lagrangian particle tracking model. Additionally, we discuss the position of our sampling relative to the buoyant and non-buoyant plumes from the hydrothermal vent.*

References

Frenger, I., Münnich, M., Gruber, N., Knutti, R., 2015. Southern Ocean eddy phenomenology. *Journal of Geophysical Research: Oceans*, 120(11): 7413-7449.

Hahm, D. et al., 2015. First hydrothermal discoveries on the Australian-Antarctic Ridge: Discharge sites, plume chemistry, and vent organisms. *Geochemistry, Geophysics, Geosystems*, 16(9): 3061-3075.

Middag, R., de Baar, H.J.W., Laan, P., Cai, P.H., van Ooijen, J.C., 2011. Dissolved manganese in the Atlantic sector of the Southern Ocean. *Deep Sea Research Part II: Topical Studies in Oceanography*, 58(25): 2661-2677.

Rintoul, S.R., 2018. The global influence of localized dynamics in the Southern Ocean. *Nature*, 558(7709): 209-218.

Rintoul, S.R. et al., 2014. Antarctic Circumpolar Current transport and barotropic transition at Macquarie Ridge. *Geophysical Research Letters*, 41(20): 7254-7261.

Schuur, C.L. et al., 1998. Sedimentary regimes at the Macquarie Ridge Complex: Interaction of Southern Ocean circulation and plate boundary bathymetry. *Paleoceanography*, 13(6): 646-670.

Sunda, W.G., Huntsman, S.A., Harvey, G.R., 1983. Photoreduction of manganese oxides in seawater and its geochemical and biological implications. *Nature*, 301(5897): 234-236.

Reviewer #2 (Remarks to the Author):

This is very well written paper looking at the iron biogeochemistry underpinning a persistent phytoplankton bloom in the Southern Ocean. The study focusses on a persistent bloom in the Pacific sector of the Southern Ocean and demonstrates that the bloom results from a combination of factors consisting of elevated subsurface iron concentrations, increased diffusivity and a bloom dominated by an efficient user of iron. The consistent occurrence of the bloom then supports further productivity at higher trophic levels and makes it an important feature in the Southern Ocean.

The authors link the elevated subsurface Fe to hydrothermal inputs from the Australian Antarctic Ridge. The novelty of the work is in the identification of further persistent blooms supported via hydrothermal inputs, and a consequent strengthening of the link between hydrothermal inputs of iron and surface productivity. This work is not the first to identify hydrothermal sources of iron as potentially important for productivity in the Southern Ocean. However, the importance of hydrothermal sources of iron to surface productivity is still a little controversial. This study is the first to use in-situ dissolved trace metal data to explore the link.

I want to emphasize here that I agree that the data support the role of increased iron fluxes from below the ferricline in fuelling the enhanced productivity in the region. The higher diffusivity, low geostrophic currents and potential upwelling combined with the extremely efficient iron use efficiency for *Pheocystis antarctica* thus seem to be very important for maintenance of the bloom and further suggest that an interesting interaction between physical and biological factors is at play. In my opinion this is already a significant and important observation.

The authors go on to use depth profiles of dissolved Mn and Fe to support the link to a hydrothermal source, ruling out ice (although the bloom starts close to the ice according to figure S1) and other sources in the process. Unfortunately, I am not convinced that the Mn data fully supports the hydrothermal source hypothesis. Indeed, Mn concentrations are enhanced below 90 m, but this is a relatively shallow maximum that is also observed in other regions of the Southern Ocean (e.g. the GIPY05 Mn transect shows that both hydrothermal influenced Mn profiles and non hydrothermal influenced Mn profiles produce the same kind of subsurface maxima). So I am struggling to really link this maximum to a hydrothermal source just 168 km away at >2000 m depth. There is clearly still a maximum in the Mn profile outside the bloom area - albeit not so pronounced (but at the same depth which would be strange if this feature was really linked to hydrothermal sources?). Furthermore there is no difference in Mn concentrations at depth. To me the Mn profiles are suggestive of an enhanced remineralisation/scavenging mechanism likely driven by the higher productivity. So overall I do not think Mn is supporting the hypothesis, and potentially even pointing to a different mechanism for sustaining iron in this region.

- *If we were seeing substantial remineralization, we would expect to see an associated oxygen depletion signal, however we see the opposite in the subsurface waters of the bloom. We actually see an anomalous water mass with an elevated oxygen signal, suggesting that we are not seeing substantial remineralization. Furthermore, we did not see any evidence of elevated particulate concentration in the CTD beam transmission data. We have not added any discussion of remineralization.*
- *We have opted to remove the DMn data and support our argument for the hydrothermal origin of the elevated DFe supporting the bloom using different lines of evidence. This is detailed in the “Case for hydrothermal DFe” section of the Discussion.*

Furthermore station 133 is presented as grey and thus outside the bloom, but it looks like it is from inside the bloom on the map given in figure 1d.

- *The depth-integrated chlorophyll for station 133 (visible in Fig. 3c) places this station on the margin of the bloom. The satellite chlorophyll in Fig. 1d does make it look as though this station might be in the bloom. This chlorophyll image is a composite of Nov 2013 through Feb 2014 and does not necessarily represent the conditions we encountered when we were on site in January 2014. If you look at the pCO₂ data shown on the cruise track of Fig. 1d, you can see that the pCO₂ is declining as you reach station 133, similarly to station 120, which is also on the margin of the bloom. We have added text to the*

caption of Fig. 1 to explain the discrepancies between the composite chlorophyll image and our in situ sampling.

Given my concerns raised above, although I do find the study technically sound and important for the field, I need more evidence with respect to the requirement for a hydrothermal source for explaining the persistence of this bloom.

- *We reworked the entire argument for hydrothermal iron and included several additional lines of evidence. This is described at the beginning of this document.*

Specific comments:

Line 62: How deep are the vents in the region and close to the bottom were the 2000 m casts?

- *The depth of the vents (~1800-2000 m) is mentioned in the discussion (L 480)*
- *We have added the station bottom depth to the description of the study region in the methods: “Bottom depth at our sampling stations ranged from 2762 m at station 135 to 3126 m at station 140/149 (Figs. 1c and d).”*

Line 153: Figure 1 panels are referred to out of order and Fig 1a not referred to at all as far as I could tell.

- *Figures 1a and 1b are now referred to in the introduction (L49 and L50)*
- *We have changed the order that we refer to figures 1c and 1d so that they are referenced in order. (L197)*

Line 178-182: nutrients not completely drawn down – so iron still limiting?

- *Iron is still very much limiting growth in this bloom. We make the point throughout the paper that it is enhanced iron delivery through the summer that allows for the bloom to persist. We have not made any changes to explicitly address this comment.*

Line 194: What was the mixed layer depth?

- *MLD has been added to the results section.*
- *The added text: “MLD showed a wide range across all sampling stations and we found no relationship between MLD and Chl a or POC concentration in the upper mixed layer. MLD at stations inside the bloom (stations 130/151, 131, and 150) ranged from 23-30 m, outside the bloom (stations 119 and 140/149) from 16-35 m, and on the margin of the bloom (stations 120, 133, 135, and 136) from 17-41 m.”*

Line 202/figure 3 caption: vertical diffusivity/K – use vertical diffusivity in caption of fig 3 too.

- *Made change in figure caption*

Line 218: So actually the difference here is linked to the species growing in the bloom and their high iron use efficiency? Why do the authors think it is Phaeocystis and not diatoms growing in the bloom?

- *Why Phaeocystis is the dominant phytoplankton (i.e. how the photophysiology of P. antarctica might account for its dominance in this particular region) is a very interesting question, but unfortunately beyond the scope of this paper. We have additional photophysiology data and iron addition bioassay experiments that we are planning to use to address this question in a subsequent paper.*

Line 261: The message I get from figure S1 is that the bloom forms close to the sea ice before the ice retreats, leaving the bloom stranded. Would remineralisation have to be unreasonably efficient in order to maintain the DFe gradient?

- *To address concerns that the AAR bloom may be an ice edge bloom, we have added Supplemental Figure S2 which shows the position of the bloom each year with mean chlorophyll concentration from October through March. The bloom clearly forms in the same position year after year, which is a clear indication of a point source of iron and is not consistent with an ice edge bloom.*

- *We do not explicitly address the idea of remineralization, but we have added information on oxygen concentration. We found that the elevated subsurface iron concentrations at bloom stations were associated with a water mass that was colder, fresher, and more oxygenated than the water mass in the same density range at non-bloom stations. The fact that this water mass has a higher oxygen concentration than waters in the same depth/density range at non-bloom stations would suggest that remineralization is not a significant component of this system, as we would expect remineralization to result in oxygen depletion.*

Line 274-281. The enhanced diffusivity is key to maintaining the supply of Fe to the surface. Reasons behind the high diffusivity are not mentioned here, but I think they should be.

- *We added two paragraphs to the results section, the first is at the end of the Enhanced DFe delivery section (L356-361) and the other is in the Case for hydrothermal DFe section (L470-476) where we outline a possible mechanism for the origin of the elevated diffusivity.*

Line 284. The potential possibly? Remove one or the other.

- *. Made this change.*

Line 307. Productivity enhanced by upwelling is commonly associated with enhanced remineralisation. Maybe the Southern Ocean is different, but was this scenario considered as a potential mechanism for resupplying remineralised Fe to the surface?

- *We do not explicitly address the idea of remineralization, but we have added information on oxygen concentration. We found that the elevated subsurface iron concentrations at bloom stations were associated with a water mass that was colder, fresher, and more oxygenated than the water mass in the same density range at non-bloom stations. The fact that this water mass has a higher oxygen concentration than waters in the same depth/density range at non-bloom stations would suggest that remineralization is not a significant component of this system, as we would expect remineralization to result in oxygen depletion.*

Line 347-349. But Mn is not elevated in deeper waters compared to surrounding waters (FigS2).

- *We have removed the DMn data from the paper.*

Reviewer 3:

Review of Schine et al

This is a very nice paper documenting a recurring area with a large phytoplankton bloom near the southern boundary of the Antarctic circumpolar current. This is an important area for carbon export from the global ocean where productivity is often regulated by Fe availability. It is also an important feeding area for marine animals. These authors suggest that this Fe is provided by hydrothermal activity on the nearby Australian-Antarctic Ridge. This is a very important finding. The provision of Fe by hydrothermal sources is an open question in the Southern Ocean and this paper does a very nice job of documenting how this might look and what the impacts might be. I find this to be a fascinating and important paper. **Hydrothermal Fe is a persistent source over time, while ice cover and other sources may have long and short term variability.** Even then, the question of how the Fe gets to the surface ocean from hydrothermal vents is an important one. This transport may be impacted by deep water formation due to changes in climate. This is a very important contribution to the scientific literature.

While I like the manuscript as a whole, I have many issues with the work as presented, but they are all addressable. So, while the bulk of my review may come across as negative (and long), please do not allow those comments to obscure my enthusiasm for the ultimate publication of this manuscript. I strongly recommend publication of this manuscript but only after significant modifications are made.

The following are places where comments about data interpretation that I find most troubling:

1. The calculations of Kz and thus vertical Fe flux are very important parts of this manuscript. In these calculations you make many assumptions. In the end, you calculate that that it is much larger than elsewhere in the southern ocean. I address the use of Kz below in my extended comments below. Those comments must be addressed. The error and error bars must be addressed and added to figure 3c.

- *This is addressed in the more detailed comments below*

2. You must define depth integrated. Over what depth interval? This is very important when you discuss ratios of Fe to other things. If you integrate the Fe over the full water column, you would get one answer and if you did it over the upper 100 m you would get another answer. For POC and chlorophyll, are you integrating from 0 to 50m or to 100m? If you integrated both POC and chlorophyll and Fe to 50 m then the ratios would be one thing and if you did all of them to 100 or 150m those ratios would be very different. This also plays into your indiscriminate use of the concept of “mixed layer depth”. What was the mixed layer depth during this study? Is it 40 m or 100m or 150m? You suggest late in the manuscript that the deepening of the MLD results in decreased productivity. I assume that you are thinking of below 100M? So, what is your mixed layer and how does it relate to all of the above?

- *We integrated Chl to 100 m and have added text describing why we chose the integration depth of 100 m to the methods. (L102)*
- *We did not calculate integrated POC or integrated DFe (we only use concentration values), and have clarified this in the methods section where we describe the calculations we make using POC and DFe (L120)*
- *We have added a description of the depth integration for NPP to the methods. (L 111)*

3. I believe that there is a significant problem with figure 3c and the associated statistics. The data density between the different profiles is significant especially for stations 120 and 136. I can imagine that if the authors made use of error bars that the error for those two stations should be rather large. The authors did not provide their data (the link did not work), so I am left to do evaluations by eye. In my imagination, most of the POC and Chlorophyll are in the upper 40 m, which means that integrated values at 120 and 136 are each defined by a single data point. May be there is a lot of POC and Chlorophyll at depth. Stations 120 and 136 do not appear in figure 3a and b, but do in c. In the end, I don't think that these data support the use of a regression line on an x-y plot. This plot, if retained must have x and y error bars. I address this further below in more detail.

- *This is addressed in the more detailed comments below*

4. The manuscript makes comparison of various rates and processes in the AAR plume versus elsewhere in the Southern Ocean. Are you comparing the biggest values in each of the systems (AAR vs Crozet and Kergeulen) or are you cherry picking the “off-shelf” values. How are you selecting off-shelf versus on-shelf? Are you always picking the off-shelf values? I'm not sure how you want to parse this, but by the end of the paper it is very unclear what you are comparing to what. I think that these elevated values out in the middle of ocean (AAR) are pretty important, however how they compare to other blooms is important as well. But how those comparisons are being made need to be clear.

- *The language that we used was confusing, and we have altered to clarify things. We were comparing the AAR bloom to other blooms in the ACC, and yes we are comparing the highest values or ranges of values. Off-shelf was only meant to differentiate from continental shelf blooms. You can find these changes in the “AAR bloom in context” section of the Discussion.*

5. There are many very general terms in this manuscript that need to be more scientific. The words “almost” and “most” should be quantified. The manuscript makes extensive use of the following undefined terms “off-shelf,” “depth integrated,” and “mixed layer depth (MLD).” Each of these concepts are central to the manuscript and must be defined, as in what is the depth range for integration, what is the actual MLD?

- *Mixed layer depth is defined in the methods as the depth of the maximum buoyancy frequency. We have removed the term “off-shelf” from the manuscript. We have defined all depth integrations in the methods as being to 100 m. We have added MLD values to the results. We have addressed vague language of “almost” in reference to how frequently the bloom appears. We have done this in the results and discussion, but still use “almost” once in the introduction as we are setting up the story and feel that the vague language is appropriate since it describes the impetus behind looking at the bloom rather than results.*

6. Data must be made available.

- *We apologize for the lack of data availability. We conflated two of the numbers in the web address for the data archive. We’ve fixed the error. The web address in the data availability statement now works properly.*

7. Figures, either text or supplement need to do a better job of showing depth resolved Fe and Mn.

- *We have removed the DMn from the paper.*
- *We show DFe as a section, depth profile, and density profile in the paper now. We have also fixed the data availability issue.*

The comments below address the ideas discussed above in more detail and also addresses other areas of important consideration.

L112 this is a very important section in the paper because much of the paper depends on this calculation. You need to tell us how Kz was calculated other than a method name and a reference. What data did you use; what are the most important parameters/variables in this calculation? In figure 3B the Kz values look discrete but the CTD data were collected at 24 Hz. Why is this? Are the data binned? From what depth did you choose your Kz for the vertical Fe flux? L118 what is the “same” depth range? You are providing almost no information to your reader and your results are not reproducible without more information here or in the supplement. If your readers are to understand what the divers are for the large Kz then you need to tell them what those variables are and how they are used. My examination suggests a rather large error for Kz and thus for vertical Fe flux. You make a significant investment in this number and you tell us that it is much larger than elsewhere in the southern ocean. Why is this? Help us understand. Maybe the primary equations from reference 26 can be shown in the supplement?

- *We have added a section to the Supplemental Material (Vertical turbulent eddy diffusivity estimates using CTD-data) that details the vertical diffusivity calculations.*
- *We added language to “Vertical DFe Flux” section in the Methods to clarify how the vertical iron flux was calculated.*
- *We added two paragraphs to the results section, the first is at the end of the Enhance DFe delivery section (L356-361) and the other is in the Case for hydrothermal DFe section (L470-476) where we outline a mechanism for the origin of the elevated diffusivity.*

For Fe gradient why did you pick the depths that you picked? Elsewhere in this review I ask about your MLDs. That information might help us understand some of your rationales for Kz and vertical Fe flux. I have trouble appreciating how the lowest Fe value in the upper 50 meters has any relationship to the highest values between 150 and 200m. I think that the MLD plays a role here. You must show x and Y error bars in figure 3c.

- *We changed the language to clarify that we are calculating the DFe flux over the ferricline. The choice of the highest deep value and the lowest shallow value is meant to quantify the strength of the iron gradient in the ferricline. See the “Vertical DFe Flux” section of the Methods.*
- *We did not add error bars to figure 3c. The error on the depth-integrated Chl a measurements was around 0.1% and error bars did not show beyond the symbol used to indicate the value. Error is not generally reported for Vertical DFe flux measurements where the Kz is calculated from Thorpe-scale analysis (see Blain et al. 2007, Nature).*

L158 what were the speeds of those currents? Do we have a sense of the residence time of the water in the bloom area? If it is three months, then perhaps you need to work on your explanations, if it is 3 weeks, then this strongly supports your suppositions and conclusions. Has the bloom also been transported in this direction? I guess that we can get a sense of this from figure S1? This is important from my perspective because you are not immediately above the ridge crest and your deep Mn does not support your shallow (see comment below). Figure S1 It would be pretty nice to see this for another year of ice break-up to demonstrate your point that ice is not the Fe source.

- *A small arrow indicating current speed is included in Fig. 1c, but we concede that this is difficult to interpret and we have added results from a Lagrangian particle tracking model (Supplemental Figure S3) that give a clear picture of the speed/direction of the currents as well as the residence time of water around both vent systems.*
- *We have added chlorophyll figures for each year in the supplemental figures (Fig. S2). They show the position of the bloom each year in the same location.*
- *We have also added a description of our sampling relative to the buoyant and nonbuoyant hydrothermal plumes in “The case for hydrothermal DFe” section of the Discussion.*
- *We have removed the DMn data from the paper.*

L161-175 Great section. Lots of very strong data. However you need to clarify what you mean by inside and outside the bloom area. In general you are consistent but line 167 suddenly discusses the edge of the bloom versus inside and outside. In fact, it is not until L 174 that we get an idea that 119 and 140 are outside. Maybe this needs to be stated at the start of this paragraph?

- *Clarified in the text which stations are inside the bloom, on the edge of the bloom, and outside of the bloom (L207)*

L168 and 171 “depth integrated” Please define the depth range for depth integration. You may need to address the concept that most of the PC and Chlorophyll are shallower than 50 m if your integration goes deeper in water column

- *The depth-integration for Chl (100 m) is described in the methods. We have added text to the methods to explain the reasoning behind our choice of integration depth. (L102)*

L172 and 173 It seems that biomass is an arbitrary unit in this case? IS it solely from the satellite data? Are units needed on biomass, did you measure it? Would chlorophyll be a better word? Forgive me, I’m no biologist, but the introduction of a seemingly unquantified variable in a well quantified section seems odd to me.

- *We used the word biomass to indicate chlorophyll and POC concentrations or integrated chlorophyll values. We changed the language to be more specific in some places (L216), but kept the term biomass in others.*

L184 any insight why *Phaeocystis antarctica* is dominant here?

- *Great question! *Phaeocystis* and diatoms have many photophysiological differences. In many cases these are related to their respective iron requirements. We chose not to go into this particular question in this paper, because it is a big one and is outside the scope of this paper. We have phytoplankton photophysiology data and results from DFe addition bioassay experiments that we plan to use to look at this question in a subsequent paper.*

L198-201 Please present Mn data in the same way that you present the Fe data so that we can see the Mn in the shallow ocean. Please provide insight on why Mn is depleted in the surface ocean. It would seem that once the Fe was taken up that there would be a relic Mn signal. Where is that Mn? Does this point to Mn limitation or any such thing? Unfortunately the link to the data did not work, so I wasn’t able to appreciate the Mn data in all of its glory. Also, while upwelling Fe might support the elevated subsurface Fe levels, this does not work for

Mn. You must discuss why this is. Is it because the surface water has transported the upwelled Fe and Mn away from ridge crest and that you are not sitting directly over the zone of upwelling Fe and Mn? Is it a water mass question?

- *We have removed the DMn data from this paper.*

L202-204 I am probably missing something here. All of your productivity is in the upper 40m, so do we need to discuss how we get the Fe into that depth range? Or is it sufficient to bring it up to 90 m?

- *We changed the text to clarify exactly how the DFe flux calculations were made in the “Vertical DFe Flux” section of the Methods section. We calculated DFe delivery across the ferricline, which would bring the iron into range of the bottom of the mixed-layer.*
- *I refer to the DFe concentration in subsurface waters (below the ferricline) because these concentrations really drive the strength of iron gradient, and also because this is where we see the most pronounced difference in DFe concentration between bloom stations and non-bloom stations.*

What is the mixed layer depth? Nitrate, Chlorophyll and POC all suggest 50m.

- *Based on buoyancy frequency estimates of MLD we had a wide range (16-41 m). This information has been added to the text (L226).*

Please provide +/- on vertical diffusivity.

- *We do not have error estimates for individual depth-binned vertical diffusivity calculations. We have added an extensive description of the K_z calculations to the supplementary material.*

Also, why is this true? Why is vertical diffusivity inside the bloom higher than outside? This is not inherently obvious. I understand that the Fe gradient would drive a higher Fe flux but the increase in vertical diffusivity needs to be explained.

- *We added two paragraphs to the results section, the first is at the end of the Enhance DFe delivery section (L356-361) and the other is in the Case for hydrothermal DFe section (L470-476) where we outline a theory of the origin of the elevated diffusivity.*

L206-219 Again, I am obviously missing something. All of the productivity is in the upper 40 meters but you are calculating Fe uptake down to 150m. This seems like a huge problem. When I integrate chlorophyll or POC, then the numbers won't change beyond 40 m??? However you integrate the missing Fe over 150 m. This really changes all of your calculations. How is this biomass removing Fe from deeper in the ocean? You really need to help your reader understand why this approach is valid.

- *We did this using concentrations, not integrated values. We have altered the language to make it clear that we are using concentration data, not integrated measurements (L257-269).*

L225 this is somewhat less than the number quoted on L216 for diatoms. Somehow these two sets of ratios (inside and outside) are much closer than for POC than they were in section above. IS this important? What does it reflect? Overall, I have trouble appreciating the importance of this paragraph.

- *The NPP values from the stations outside of the bloom changed substantially due to an error in our calculations. The Fe:C values outside the bloom are no longer near the diatom range. (L271)*
- *We changed the language of the paragraph to better express the importance of this particular analysis (L266). Essentially it's a balance between Fe delivery and Fe uptake (which is expressed as the Fe required to balance the amount of carbon we know is being taken up in the ML. What we show is that the Fe requirement dictated by the NPP measured in the bloom can be entirely explained by the Fe delivery into the mixed-layer.*

L230-232 Vertical Fe flux is a function of both vertical diffusivity and the Fe-gradient. So why doesn't integrated chlorophyll correlate with all three variables? Something is strange here. Again, why is this important, and what does it mean?

- *We apologize for the confusion. In short, high K_z with no iron would not produce a bloom, neither would high iron without some way of getting it to the surface where it could relieve iron limitation. You have to have both the limiting nutrient and the mechanism for delivering it. The fact that both the iron gradient and the vertical diffusivity are required is the point.*

Where are the data from station 136; they don't show up in figure 3 a and b but do in c? Same is true for stations 133 and 120.

- *Figures 3a and 3b would be too busy with all of the profiles, so only the stations with the highest and lowest biomass are shown. We are showing the two extremes to illuminate the difference.*

Stn 133 doesn't even show up in Figure 2. In Figure 3c, 136 (and 120) appears to be an outlier? Fe gradient looks smaller for 136 but larger for 120, what about the diffusivity?

- *Wanted to make a section that showed a clear gradient from outside to inside the bloom, Station 133 was at a longitude that would put in the middle of the section, but the station was further north than the bloom extended. We opted not to include Station 133 in the sections as it disrupted the bloom portion of the section and made it harder to see the difference between bloom stations and non-bloom stations and the very quick transition between them. We did find it valuable to include data from station 133 in other sections of the paper, because it provides important information about the bloom margin and we only had a few stations worth of data on the margin of the bloom.*
- *We were unable to include profiles from every station in our profile figures due to issues of clarity. Therefore, we only included stations with undeniably high biomass and undeniably low biomass, to illustrate the differences between these two end members. In fixing the data access issue, we hope that readers will be able to use the full data set to answer lingering questions.*

Integrated chlorophyll is based on a single data point for 136 and 120? Hard to tell what the numbers are because your data set is not available and ODV is well known to create data like it has for the shallow chlorophyll at these stations. Is it reasonable to estimate the integrated chlorophyll without the shallower data points? Again, I don't have the data set but ODV suggests that the deeper ones are largely irrelevant? So, again, I ask how did you do this integration?

- *For some reason ODV was excluding several surface data points. We have fixed this issue. The chlorophyll integration was not done in ODV and those numbers remain the same. Also, we have fixed the data availability issue.*
- *The chlorophyll integration is described in the methods section (L 102-105).*

Does station 136 belong in your figure 3c statistical analysis. I can guarantee you that if you remove this data point that your correlation would improve. The same point should be made about station 120 and 133. In the end, the statistical validity of this entire section seems to be dubious at best. You really need to rethink this. What you are saying may be true. The increase in chlorophyll (and POC?) with increased Fe flux seems reasonable, but your data may or may not support the use of stations 120, 133, and 136 in this analysis. I suspect a table would be more helpful than this graph. You have two overlapping data clusters, not a trend.

- *Excluding the intermediate stations would not allow for enough remaining stations to conduct either a regression analysis or a t-test comparison of the bloom vs. non-bloom stations. You're absolutely correct that the relationship would improve with the exclusion of certain stations, however we do not have grounds to exclude them. We stand by the statistics of this figure, with the caveat that it is only meant to illustrate a very simple and straightforward relationship between DFe flux and integrated chlorophyll.*

L242, 247 Doesn't most of the Fe from Kerguelen ultimately come from the shelf? Are you talking about proximity to the shelves? Are you dismissing the shelf-blooms? Are you instead wanting to say that your blooms are the largest in the open ocean away from shallow topography and margin/shelf influence? I can see this importance, but it seems odd putting it in terms of a negative feature (non-shelf). I suggest that you refer to

this in a more positive sense. Shelves are important. I understand what that you want to say that this bloom is not near any shelf. I would think about how to present this. It's not a deal breaker, but it was hard to figure out what you were wanting to say. To this reader it was very distracting to understand what your backwards logic was trying to express.

- *We just meant to exclude shelf blooms like the one that forms each year in the Ross Sea. We altered the language to refer to ACC blooms rather than shelf and non-shelf blooms. These changes are in the "AAR bloom in context" section of the Discussion.*

L249 How do your outside the bloom values compare to the rest of the SO?

- *The production rates measured outside the bloom are lower than the satellite-based production estimates for the height of the spring bloom for the entire SO. The values we measured for in situ production outside of the bloom are stated in the results section. We have added satellite-based daily production values to the text so that readers can make their own comparison.*
- *The new text reads: "Our daily NPP measurements in the bloom (1.10-1.14 g C m⁻² d⁻¹) are 2.7-3.8 times higher than the satellite-based daily NPP estimates at the peak of the spring bloom in open SO waters (0.30-0.40 g C m⁻² d⁻¹)"*

L250 What part of Crozet? What depth? Are there shelf and non-shelf values at Crozet that you are comparing to? This whole section really makes it difficult to know what is being compared. At this point I think that you might need a table!

- *We changed the language to clarify what is being compared (L322). These are a range of values from all over the Crozet bloom. Additional information can be found in the specific reference. As to the question of depth, we're not sure exactly what you mean, but NPP is generally integrated down to the 1% light level. I checked and that is what was done for the study referenced here, which is the same method that we use, so the numbers are comparable.*

L272, you MUST say why your Kzs are higher than elsewhere. What is the physical reason? IS it knowable? This is a great reason to show this calculation somewhere so that we can have faith that this order of magnitude is real. Gut instinct of your reader is that you made a mistake. Because this is a major major part of your story, you must make it so that the reader can understand it.

- *We added two paragraphs to the results section, the first is at the end of the Enhance DFe delivery section (L356-361) and the other is in the Case for hydrothermal DFe section (L470-476) where we outline a theory of the origin of the elevated diffusivity.*

The Case for hydrothermal DFe (L 328)

You might want to consider Tagliabue and Resing 2016 in this set of arguments. They seem to show that the isopycnals crossing this section of the ridge crest are similar to those that are also upwelling in your study area. They also make arguments about the importance of hydrothermalism in the southern ocean versus elsewhere.

- *We have looked at Tagliabue and Resing 2016 and find that it makes some excellent points about the impact of hydrothermalism in the Southern Ocean, but we did not add this reference. Our understanding of the main findings of Tagliabue and Resing (2016) is that they are estimating the contribution of hydrothermalism in the different ocean basins (including hydrothermalism in the Southern Ocean itself) to the overall iron inventory in the deep Southern Ocean and subsequently the impact on carbon export. They also look at the impact of iron stabilization by ligands on the impact of hydrothermal Fe on carbon export in the Southern Ocean. This paper is looking at very large-scale issues that we never speak to in our paper as we are focused on a very particular region as opposed to the entire Southern Ocean. We chose not to include this reference.*

While your deeper Fe supports your shallower Fe, the same is not the case for Mn. Is this because you are not in the "stem" of the up welling? Perhaps Fe and Mn are upwelling upstream and the surface waters move in a different direction at a different speed than in the deeper waters? Can you address the dMn minimum in the

surface ocean? Can the hydrothermal Fe and Mn accumulate under the ice during the winter? You need to make a case for there being more Mn in the near sub surface than at depth. I don't think that it can be light driven recycling. Mn is typically low in the southern ocean and deep mixing will ultimately mix out the Mn on a seasonal basis. Perhaps I am incorrectly assuming that the residence time of Mn in this region is short compared to the rest of the ocean.

- *We have added text describing hydrothermal plume dynamics and clarify that we are sampling outside of the buoyant plume or “stem” of the upwelling. We are using this to put the DFe profiles in a different context, explaining that we would therefore expect to see a decrease in the disparity between bloom and non-bloom profiles below the depth of the nonbuoyant plume.*
- *Because a detailed discussion of DMn data was outside the scope of this paper but was called for by multiple reviewers, we have opted to remove our DMn data from the paper, and rather support the argument for the hydrothermal origin of the DFe that drives the bloom with other lines of evidence. This evidence is laid out in detail in the “Case for Hydrothermal Iron” section of the Discussion. The text describing the buoyant and nonbuoyant plume dynamics is also in this section.*

L383-386 Here is the whole shelf/ non-shelf set of questions again? Are you only comparing AAR to the lower off-shelf values at Crozet and Kerguelen? I find myself to be very very confused about this issue. Do you have a sense of the total flux (rate x area x time length) or total flux per m² (rate x time length) for the different systems?

- *These are the shelf values for the Ross Sea bloom and the bloom that forms downstream of the Kerguelen plateau. We have taken the shelf/off-shelf distinction out of the other section of the paper, which should help readers to understand what we're talking about here (since we haven't confused them in a previous section of the paper). In this section we are referring to the mean CO₂ fluxes measured in the Kerguelen and Ross Sea blooms, no shelf/off-shelf distinctions. We have not changed any of the language here, because we feel that without the previous confusion this section will make sense.*
- *The strength of this bloom as a CO₂ sink is a question for another paper. We are only indicating the potential at this point. Digging into this question is beyond the scope of this paper. But it is definitely an interesting direction for subsequent work.*

L 387-396 I like this section. I thought it was gratuitous when I first read it. However upon contemplation I realize that climate change could weaken both down and upwelling, decreasing Fe input and the amount of forage available.

I guess I would like to see the conclusions address the hydrothermal input and the unique situation of the front over the ridge crest. I would like it to address the concept that hydrothermal sources are persistent while other sources may experience seasonal, annual, decadal, and longer variability. On the one hand the persistent source is important, on the other, if it can't get to the surface ocean, then it is less so.

- *We have reworked the Implications section to address the idea of hydrothermalism as a consistent source of iron in the final paragraph of the paper.*

Details to be addressed:

Abstract

The word unprecedented means “never done or known before.” You proceed in the paper to document several regions in the southern ocean where similar observations of recurring blooms in specific geographic areas. You might want to rethink the use of this word. You don't need this to convince me that your observations are important.

L13 later in the paper it is “-32” The units should be in mg d⁻¹ m⁻², I hope.

- *Changed*

L33 Ardyna et al used surface drifters in conjunction with historical 3-He data. 3-He was not measured as a part of their study. Your sentence needs to better reflect this.

- *Changed language*

L37 Do you accept their hypothesis of enhanced eddy kinetic energy? Should that apply here? Why don't you calculate it?

- *We added EKE data from Argo floats (Figure 4 and Supplemental Figure S3), but we don't really discuss it too much since this region is marked by very low EKE.*

L47 If the records exist, and you do reference them, can you need to quantify "almost". Almost is not a great word for a scientific paper.

- *We have added figures of mean chlorophyll from October-March for each growing season from 1997-2019 to show exactly how many years out of the last 22 years the bloom is present. We do not mention this in the introduction, but we do mention specific numbers in the Discussion (L312).*

L82 what do you mean when you say "Samples were digested for 4 h....." please provide a reference or a description.

- *We have removed the DMn data and therefore the sampling and analysis methods from the paper.*

L84 Did you distill reagent grade acid? How did you do it? Was it a sub-boiling distillation or a 2x quartz distillation? Perhaps you purchased the acid.

- *We have removed the DMn data and therefore the sampling and analysis methods from the paper.*

L86 do you mean a serial dilution of a 1000ppm standard? It seems very difficult to use a 1000ppm standard to spike your samples. If it was a purchased standard, can you provide the manufacturer?

- *We have removed the DMn data and therefore the sampling and analysis methods from the paper.*

L134 please define SSM/I

- *Made change*

L138-139 It would seem that a single Arrigo reference would suffice.

- *The first Arrigo reference details the algorithm, the second reference describes the current data products and exact methods used. We can see where this might seem excessive, but they provide different information necessary for the reader to recreate our methods. We changed the language to indicate that the two references provide different information (L152).*

L254 to 257 Is this time scale information needed for anything other than information? It seems all three time length scales are the same, so what can the reader do with this information? Maybe it belongs in the introduction?

- *The time scale is an important component since carbon fixation is measured as a daily rate, therefore the longer the bloom persists the more CO₂ it has the potential to draw in. The fact that the AAR bloom persists for the same length of time as the Kerguelen and Crozet blooms is an indication that it is not simply a marginal ice zone bloom, which would be expected to last for a much shorter period of time. We have left this section as it was.*

L262 Really need to define the mixed layer. It keeps showing up, but with no definition.

- *We define the depth of the upper mixed layer in the methods section (L76).*

L291 please use "Advection from continental shelves...."

- *Made change*

L296 While I do agree with your assessment, I note that the ocean margins and associated slopes do go to 2000m.

- *Changed the language so that we don't mention shelf advection in the argument that it cannot explain elevated DFe down to 2000 m. IN the "Possible DFe Sources" section of the Discussion.*

L334 do you have an actual +/- value for "same location" Perhaps not important???

- *Added images of the bloom for each year (as Supplemental Figure S2) that give a clear idea of what is meant by "same location."*

L340 the referenced paper does not report dissolved Fe. It reports unfiltered total HCl leachable Fe and Mn.

- *We made sure that any references to Hahn et al. 2015 refer to Fe not DFe. The rewrite of this section made this specific comment no longer relevant.*

L372 Good point, again I would consider Tagliabue and Resing reference from above.

- *We feel that adding a reference here is unnecessary*

L378 I would use "from the advection of a shelf source,"

- *Made change*

Figures

Figure 1. IS the white in the figure ice or land? Could you please add that to the caption?

- *The white is no data, which is either land or persistent sea ice. We have added this to the figure caption.*

Figure 2 Where are data from station 133.

- *The data from station 133 are not included in the section because we move out of the bloom at that station and wound up sampling on the margin. The consensus among the authors was that we should show as continuous a slice of the bloom as possible in the sections, and so station 133 was excluded.*

Supplement should show all vertical profiles. ODV makes up data and puts features where there are none. This is happening in panel a and b, but I can tell that the gridding is more appropriate in panels c and d. The greater data density in c and d (vis a vid stations 120 and 136) allow you to more honestly grid your data. In addition, you integrate the water column, without telling us what that means. How deep did you go? Do Chlorophyll and POC go to zero? Is the upper water column extremely more important than below 40m? I can't tell. SO the addition of vertical profiles at the appropriate depth range and a definition to the depth range of integration are crucial. If there is significant chlorophyll and POC at depth, then you might consider using a stretched color scale in ODV.

- *Having fixed the data availability issue, interested parties can now plot and replot all of the data. We feel that the inclusion of depth profiles for each station for all of these variables is unnecessary with the data being publicly available.*
- *The depth integration methods for Chl and NPP are detailed in the appropriate sections of the methods.*

Figure 3a. Where are the data for stations 120, 133, and 136?

- *Including 3 additional depth profiles would make this figure unreadable. We chose to show only the endmember stations that were either unambiguously in the bloom or out of the bloom in order to illustrate the differences.*

Figure 3b. putting a line through these data is a bit of a stretch. Better to make two groups and take averages and standard deviations. While your p-value may tell you one thing, looking at the data tells me that the relationship is not strong enough for adding in a line.

- *This is only meant to illustrate a simple relationship. We do not have enough stations to do a t-test with only the bloom and non-bloom stations.*

Figure 4. IS the white in the figure ice or land? Could you please add that to the caption?

- *The white is no data, which is either land or persistent sea ice. We have added this to the figure caption.*

Figure S2 Do you have all of the data?

- *We have removed the DMn data from this paper.*

Figure S3 If these data have not been published, have you been in contact with the authors about using them in this fashion? Maybe you need to reference the International Geotraces data product? Can you put in a comparable ODV plot of Mn in the upper 400 m in the supplement?

- *We have added the Geotraces intermediate data product reference to the text of the paper (not to the figure caption).*

This review was completed by Joseph Resing

REVIEWER COMMENTS

Reviewer #1 (Remarks to the Author):

The corrections to manuscript NCOMMS-20-07481A by Schine et al. has result in a greatly-improved version. There is a no doubt that increased iron supply (relative to waters away from the Australian Antarctic Ridge system) drives the large phytoplankton bloom observed by Schine et al.. The question that I had in my first review was whether the iron supplied to the surface ocean over the Australian Antarctic Ridge system could be directly associated with the hydrothermalism. In the revised manuscript Schine et al. have presented new hydrographic measurements in Fig 3 and an increased discussion section on hydrothermal iron supply. The new panels in figure 3 and the revised discussion supports the idea of hydrothermal iron supply into the water column underlying the bloom along the Australian Antarctic Ridge.

Based on the corrections made to the revised manuscript, I think that it will make nice addition to the scientific literature.

Michael Ellwood

Reviewer #2 (Remarks to the Author):

I very much enjoyed reading the revised paper of Shine et al.,. I find the more detailed explanation of the particular physical environment that facilitates the transport of the non buoyant plume from the Australian Antarctic Ridge into the deep winter mixing zone of this particular sector of the Antarctic Circumpolar Current convincing. The authors have dealt well with my initial concerns and I think the hypothesis is well supported by the arguments. The study makes an excellent and exciting contribution to the field and I fully support publication.

I have one very minor comment regarding the caption of Figure 3: I think there maybe a typo regarding the standard deviation of the DFe measurements.

Reviewer #3 (Remarks to the Author):

I really think that this is a great paper reporting on a bloom in the southern ocean that reappears virtually every year in roughly the same spot. The cause of the bloom must be the elevated iron just below the surface ocean. This iron is transported to the surface ocean due to vertical diffusivity. The question is where does the Fe come from.

The in-depth analysis of different components of the biological system in the surface ocean are very nice. In addition the transfer of elevated Fe from just below the surface to the surface ocean seems reasonable to me, however I have not delved into the concepts and the math of the vertical diffusivity, but there is a long well referenced section in the supplement. For better or for worse, the authors choose to make the crux of the paper about a singular source of Fe to the surface ocean saying "...we contend that it is the increased Fe inventory in the upwelled water resulting from hydrothermal vent emissions that stimulates and sustains the AAR bloom each year." Here they use a variety of arguments. The authors rely heavily on "particles" released into an unnamed undocumented model. It is unclear from the manuscript if the particles reach the surface ocean. I discuss the particle tracking in detail below. The authors also rely heavily on hydrographic arguments. The basic idea of the isopycnals tilting upward from the deep ocean and transporting water upward from the depths of the ridges is very reasonable and is highly suggestive that hydrothermal Fe must play a role in surface productivity. I agree with this. However the arguments that are made on behalf of this hypothesis are

troubling; with the most important problem being that the hydrography doesn't appear to fully support their claims. I discuss the hydrography much more extensively below. Finally, the elevated Fe deeper in the water column at stations 150 and 151 versus at 149 also suggest a deep Fe source in this region. So, while I think it is reasonable that hydrothermal Fe is important in this region, I'm not sure the evidence used in support of this hypothesis is very robust.

I see that in the reply to reviewers that two reviewers wanted to know the source of the phaeocystis. This seems like a good question, as it too may indicate where (some of) the water is coming from. Given all the added text, it seems odd this is omitted. It is disappointing that the authors don't have a short answer to satisfy the readers of the manuscript. From the Occam's Razor perspective, the surface currents are from the south (Figure 1), there are islands and their banks to the south, there is a huge shelf to the south, and maybe the Phaeocystis comes from the south. Figure 1c, strongly suggests that the currents from the south converge with those from the west and North west at the location of the bloom. From the simplest perspective, doesn't this strongly suggest at least some of the water comes from the south. The ice edge is reasonably close by. Young Island is only 300 or so km from the bloom while the ridge is around 200km. There is also a substantial shelf at 500m depth 500km away that abuts the continent. In addition, as I discuss further below, the hydrography supports a southern source. Did you do particle tracking from the ice edge, from the islands, or from this continental shelf??

My conclusion is that this is, in general, a great manuscript. The reoccurring bloom for 20 years is very compelling. The transfer of the Fe from just below the surface ocean is also pretty impressive. Unfortunately I don't think that there is enough evidence hydrothermal iron is the only source of Fe. I would love for there to be helium data to prove me wrong. I do think that the hydrothermal Fe does rise up within mixing distance of the surface ocean in this region as is supported by the elevated Fe throughout the water column in Figure 3a stations 130 and 131. I agree that emplacing a hydrothermal plume into the uprising isopycnals will put those Fe into the upper 500m. So, while I think hydrothermal Fe plays an important role, I think that the arguments presented here have some flaws that need to be addressed. I also think that new evidence presented in the revision indicates that there is a southern source for some of the Fe. I discuss these major point below.

Particle Tracking

Particle tracking seems to be an important piece of evidence in this manuscript. However it is unclear from figure S3 and the text if the particles move south east and upwards or if they just move to the South East (L298 says southwest, but I think that is an error?). If the particles only move laterally, that would be important to know. I don't quite have time to wade through Sergi et al., but I'm guessing they track the particles to the surface? It is unclear how you completed the particle tracking. Did you ingest their model or is there a web tool. I see that Sergi is now a co-author, however it is important to document the methodology. For how many years did you repeat the particle release? I assume that the driving forces change annually or are they based on some fixed climatology. In figure 1c or d, a distance bar would enable your reader to know how far the bloom is from the ridge crest? This is important because your 50km disk for particle release is fairly large compared to the features on your map. If you look carefully at the figures in Hahn et al you can also fairly infer that the plumes are drifting to both the South and the East in the deep ocean, which provides some additional support to your arguments.

The currents in figure 1c suggest that the bloom is at a convergence zone where northward, eastward and southeast ward surface currents come together. This is also clearly a region of upwelling and where deep water rises towards the surface. You say that currents in the region are slow. Over what depth range are they slow, and what is slow? You also say surface currents are consistent with particle release, yet the only particle release you show is at the ridge crest so it is unclear what this means as Figure 1C suggests that the surface vectors converge at the bloom. Dotto et al have data over 5 years. Do the surface currents always converge on the bloom area? If they don't, then this would add support to your hypothesis that the Fe is exclusively hydrothermal and does not include any from the

shelf. If you track particles from the ice edge, Young Island, or closer to the Antarctic continental shelf where do they go?

Please note that it would help your readers to know how you extracted the surface currents from Dotto et al., 2018. If you look carefully at the figures in Hahm et al you can also fairly infer that the plumes are drifting to both the South and the East.

Hydrography

Hydrothermal plumes rise 100-300m off of the seafloor and injecting them straight onto sigma theta 27.81 rising from the deep pacific undoubtedly carries that hydrothermal Fe (and Mn) upwards. I like the arguments about inputting hydrothermal water into the upwelling limb and this must be a transport pathway for hydrothermal Fe to reach the surface ocean. I think that figure 4 shows this and is reasonably compelling; I can roughly reproduce it using the CLIVAR 1-degree gridded data. It is important that you identify the "data" source. I also think that your argument would be more compelling if you limited the x-axis to 130E to 180E. That will really make the sloping isopycnals more compelling. As I said, it is unclear where the data in figure 4 comes from. The figure caption says that Figure S3 shows a map of where the data come from (I think it is meant to be Figure S4), however there is still no mention of where the EKE, isopycnal pressure, and bathymetry come from. I assume that the NPP data source is discussed elsewhere but I did not check.

There are some significant problems with your suggestion that there is an observable hydrographic anomaly associated with the non-buoyant plume. It appears to me that the water at depth is not cold enough and definitely not fresh enough and probably not rich enough in oxygen to cause the deviations observed here. The biggest problem is that the salinity anomaly is in the wrong direction. Figure 3 shows a freshening; 2000m water is around 34.7 (Figure S7 and Figure 3), which is saltier than the overlying water between 27.6 to 27.8. As for the temperature deviation, the deep water is around 1C but most of the water in the density 27.6 to 27.8 interval is much colder. There maybe a similar problem for oxygen based on Figure 3. Unfortunately the data in figure S7 are in different units. By contrast, the water at 50 m is much colder, fresher and more oxygen rich. I feel like the only water that solves this is from the south. I made three plots of the CLIVAR 1-degree gridded data set, from GIPY6, and from a conglomeration of data sets through GLODAPV2 2020. They are posted at the end of my review. IN each plot Fresher, colder more oxygen rich water in the upper 500m are visible at the southern ends of the plotted lines. For me it seems there is a lot of evidence that some of the water comes from the south. I think you need to acknowledge this in your manuscript.

Figure 1 and S3 needs the frontal zones labeled on the map. Given the number of references to them in the paper, the reader needs to see them on the figure without rereading the caption. You need a reference for maximum buoyancy frequency. Please give density range of UCDW as this would be important to figure S6. Might also indicate UCDW on figure S6

Hydrothermal

Megaplumes (also known as event plumes) are formed episodically and thus are the antithesis of the source you want. You want long lived consistent sources. Invoking mega plumes is a mistake. There is no reason to include them in your discussion. Yes they have greater buoyancy but they are episodic and not seasonal.

Not all hydrothermal effluent is low salinity and hot fluids of normal and high salinity also rise into the water column. We know nothing about the end-member fluid composition at these sites, so I suggest that you omit this argument.

Referencing the German Treatise in Geo chemistry is pretty unfair to the people who did the work. To fully appreciate the dynamics of rising plumes would require a little more effort. For example, high temperature fluids are diluted >1000X with ambient water within 10s of meters of the vent orifice. As a result, any plume that would affect the hydrography in the upper ocean would have the imprint of that deep water. Based on a wide deep body of literature there are many reasons to be careful about claiming a hydrographic anomaly from venting.

Omitting the Mn data because you had trouble explaining it is a disappointing outcome.

Other points to consider

The data point at station 151, 1400m appears to be a mis-trip based on Fe, Nitrate, and Silicate. Figure 1A is not phytoplankton, per se, but chlorophyll.

Also, you should reach out to Andy Bowie about how he wants his data referenced. It seems really unfair to reference Schlitzer for his Fe and hydrography data. At least reference his cruise report. The data panel in ODV shows the data source.

Figure 1 Glodap V2 2020

Figure 2 Clivar 1 degree gridded

eGEOTRACES

Figure 3 GIPY6

Reviewer #3 (Remarks to the Author):

I really think that this is a great paper reporting on a bloom in the southern ocean that reappears virtually every year in roughly the same spot. The cause of the bloom must be the elevated iron just below the surface ocean. This iron is transported to the surface ocean due to vertical diffusivity. The question is where does the Fe come from.

We thank the reviewer again for their kind comments!

The in-depth analysis of different components of the biological system in the surface ocean are very nice. In addition the transfer of elevated Fe from just below the surface to the surface ocean seems reasonable to me, however I have not delved into the concepts and the math of the vertical diffusivity, but there is a long well referenced section in the supplement. For better or for worse, the authors choose to make the crux of the paper about a singular source of Fe to the surface ocean saying "...we contend that it is the increased Fe inventory in the upwelled water resulting from hydrothermal vent emissions that stimulates and sustains the AAR bloom each year." Here they use a variety of arguments. The authors rely heavily on "particles" released into an unnamed undocumented model.

The model is not unnamed and undocumented. It is based in the work of Sara Sergi (Sergi et al 2020), whom we cite in the paper.

We disagree with the statement that we have made the crux of the paper about a singular source of Fe. In the sentence from our study quoted above we clearly say that it is the addition of hydrothermal Fe to the existing Fe inventory that creates the conditions that make this recurring bloom possible. Without the background Fe concentration, the additional hydrothermal Fe might not be enough to create a bloom.

It is unclear from the manuscript if the particles reach the surface ocean. I discuss the particle tracking in detail below. The authors also rely heavily on hydrographic arguments. The basic idea of the isopycnals tilting upward from the deep ocean and transporting water upward from the depths of the ridges is very reasonable and is highly suggestive that hydrothermal Fe must play a role in surface productivity. I agree with this. However the arguments that are made on behalf of this hypothesis are troubling; with the most important problem being that the hydrography doesn't appear to fully support their claims. I discuss the hydrography much more extensively below. Finally, the elevated Fe deeper in the water column at stations 150 and 151 versus at 149 also suggest a deep Fe source in this region. So, while I think it is reasonable that hydrothermal Fe is important in this region, I'm not sure the evidence used in support of this hypothesis is very robust.

We address this below where the reviewer made more detailed comments.

I see that in the reply to reviewers that two reviewers wanted to know the source of the phaeocystis. This seems like a good question, as it too may indicate where (some of) the water is coming from. Given all the added text, it seems odd this is omitted. It is disappointing that the authors don't have a short answer to satisfy the readers of the manuscript. From the Occam's Razor perspective, the surface currents are from the south (Figure 1), there are islands and their banks to the south, there is a huge shelf to the south, and maybe the Phaeocystis comes from the south. Figure 1c, strongly suggests that the currents from the south converge with those from the west and North west at the location of the bloom. From the simplest perspective, doesn't this strongly suggests at least some of the water comes from the south. The ice edge is reasonably close by. Young Island is only 300 or so km from the bloom while the ridge is around 200km. There is also a substantial shelf at 500m depth 500km away that abuts the continent. In addition, as I discuss further below, the hydrography supports a southern source. Did you do particle tracking from the ice edge, from the islands, or from this continental shelf??

There is simply no way to know where the Phaeocystis in the AAR bloom came from. The cells that seeded the bloom may have come from the south or they may have come from somewhere else. There is no way to know for sure since this species is distributed widely throughout the Southern Ocean. The point here is that the AAR region clearly provided the conditions that were conducive for Phaeocystis to bloom.

We have altered the language on L378 to acknowledge that these currents from the south exist, but it does not change our analysis that the shelf is an unlikely source of iron in this region. This conclusion is based both on the results of the Ardyna et al. (2017, 2019) iron advection model as well as the fact that the bloom forms in the same location year after year. If the iron was moving laterally from the shelf, we would expect to see elevated production along the flow path and more variation in the interannual position of the bloom (a point we make in the paper about needing a point source of iron to explain the consistent position of the bloom).

My conclusion is that this is, in general, a great manuscript. The reoccurring bloom for 20 years is very compelling. The transfer of the Fe from just below the surface ocean is also pretty impressive. Unfortunately I don't think that there is enough evidence hydrothermal iron is the only source of Fe.

We do not contend that hydrothermal is the only Fe source. Small amounts of Fe are found in surrounding waters and support a modest phytoplankton population. However, we believe that the hydrothermal vents supply the additional Fe needed for the AAR bloom to reach the high concentrations we observed.

I would love for there to be helium data to prove me wrong. I do think that the hydrothermal Fe does rise up within mixing distance of the surface ocean in this region as is supported by the elevated Fe throughout the water column in Figure 3a stations 130 and 131. I agree that emplacing a hydrothermal plume into the uprising isopycnals will put those Fe into the upper 500m. So, while I think hydrothermal Fe plays an important role, I think that the arguments

presented here have some flaws that need to be addressed. I also think that new evidence presented in the revision indicates that there is a southern source for some of the Fe. I discuss these major point below.

We also wish that we had some ^3He data!

Particle Tracking

Particle tracking seems to be an important piece of evidence in this manuscript. However it is unclear from figure S3 and the text if the particles move south east and upwards or if they just move to the South East (L298 says southwest, but I think that is an error?).

This was an error, which we corrected. We apologize for any confusion. L299

If the particles only move laterally, that would be important to know. I don't quite have time to wade through Sergi et al., but I'm guessing they track the particles to the surface? It is unclear how you completed the particle tracking. Did you ingest their model or is there a web tool. I see that Sergi is now a co-author, however it is important to document the methodology. For how many years did you repeat the particle release? I assume that the driving forces change annually or are they based on some fixed climatology. In figure 1c or d, a distance bar would enable your reader to know how far the bloom is from the ridge crest? This is important because your 50km disk for particle release is fairly large compared to the features on your map. If you look carefully at the figures in Hahm et al you can also fairly infer that the plumes are drifting to both the South and the East in the deep ocean, which provides some additional support to your arguments.

We clarified the particle tracking methods.

We added the distance from the center of the vent to station 150 to the caption for Figure 1 to give readers an idea of the approximate scale of the bloom.

However, we cannot put a distance bar on figures 1c or 1d because of the projection used, which gives equal size to each degree of latitude and longitude, however this means that the size of a lat/lon degree box changes as you move north/south. We chose this projection to make it easy for readers to see the exact coordinates of the bloom and the vents, and because the distortion effect of this type of map projection on an area this size is minor. It does, however, prevent us from putting a precise distance bar on the map.

The currents in figure 1c suggest that the bloom is at a convergence zone where northward, eastward and southeast ward surface currents come together. This is also clearly a region of upwelling and where deep water rises towards the surface. You say that currents in the region are slow. Over what depth range are they slow, and what is slow? You also say surface currents are consistent with particle release, yet the only particle release you show is at the ridge crest so it is unclear what this means as Figure 1C suggests that the surface vectors converge at the bloom.

Dotto et al have data over 5 years. Do the surface currents always converge on the bloom area? If they don't, then this would add support to your hypothesis that the Fe is exclusively hydrothermal and does not include any from the shelf. If you track particles from the ice edge, Young Island, or closer to the Antarctic continental shelf where do they go? Please note that it would help your readers to know how you extracted the surface currents from Dotto et al., 2018. If you look carefully at the figures in Hahm et al you can also fairly infer that the plumes are drifting to both the South and the East.

Unfortunately, particles only move laterally in the model we used. This has been clarified in the description of the model in the Methods section.

We see an area of convergence to the southwest of the bloom (65 S and 165 E), but not in the region of the bloom. Furthermore, Figure 1c shows the current trajectories at one point in time. The zone of weak convergence that can be seen in this figure, to the south of our study area, actually moves significantly over time and does not correspond well with the consistent location of the AAR bloom. This has been clarified in the legend for Figure 1d.

The iron advection model from Ardyna et al. (2017, 2019) provides a comprehensive analysis of the expected advection pattern of Fe from contact with continental shelves and islands. We refer the reviewer to these papers as the authors have comprehensively answered the question of iron advection patterns based on Lagrangian particle tracking for the entire Southern Ocean. We feel that it is unnecessary to repeat the work.

The surface currents from Dotto et al. (2018) were the data published with the manuscript.

Hydrography

Hydrothermal plumes rise 100-300m off of the seafloor and injecting them straight onto sigma theta 27.81 rising from the deep pacific undoubtedly carries that hydrothermal Fe (and Mn) upwards. I like the arguments about inputting hydrothermal water into the upwelling limb and this must be a transport pathway for hydrothermal Fe to reach the surface ocean. I think that figure 4 shows this and is reasonably compelling; I can roughly reproduce it using the CLIVAR 1-degree gridded data. It is important that you identify the "data" source. I also think that your argument would be more compelling if you limited the x-axis to 130E to 180E. That will really make the sloping isopycnals more compelling. As I said, it is unclear where the data in figure 4 comes from. The figure caption says that Figure S3 shows a map of where the data come from (I think it is meant to be Figure S4), however there is still no mention of where the EKE, isopycnal pressure, and bathymetry come from. I assume that the NPP data source is discussed elsewhere but I did not check.

The source data for EKE and isopycnal pressure can be found towards the end of the Methods section in the section called "Argo-derived data products". We state where the data products come from (including links to the websites where the data can be found) as well as what they were used for in the paper. The link to the bathymetry data source was in the caption for Figure 1 and has now been added to Figures 4 and S4 where the data are also used.

We have corrected the figure caption for Figure 4 to reference Figure S4 rather than S3.

There are some significant problems with your suggestion that there is an observable hydrographic anomaly associated with the non-buoyant plume. It appears to me that the water at depth is not cold enough and definitely not fresh enough and probably not rich enough in oxygen to cause the deviations observed here. The biggest problem is that the salinity anomaly is in the wrong direction. Figure 3 shows a freshening; 2000m water is around 34.7 (Figure S7 and Figure 3), which is saltier than the overlying water between 27.6 to 27.8. As for the temperature deviation, the deep water is around 1C but most of the water in the density 27.6 to 27.8 interval is much colder. There maybe a similar problem for oxygen based on Figure 3. Unfortunately the data in figure S7 are in different units. By contrast, the water at 50 m is much colder, fresher and more oxygen rich. I feel like the only water that solves this is from the south. I made three plots of the CLIVAR 1-degree gridded data set, from GIPY6, and from a conglomeration of data sets through GLODAPV2 2020. They are posted at the end of my review. IN each plot Fresher, colder more oxygen rich water in the upper 500m are visible at the southern ends of the plotted lines. For me it seems there is a lot of evidence that some of the water comes from the south. I think you need to acknowledge this in your manuscript.

We removed the GEOTRACES data that comprised Figure S7 in our last version and have replaced it with depth profiles (down to 2000 m) of potential temperature, salinity, oxygen, and potential density anomaly from our own cruise. The stations used match those displayed in Figure 3. These depth profiles help us to demonstrate several things.

The cold tongue of water coming from the south (also fresh and oxygen-rich) identified by the reviewer, is clearly visible in our depth profiles at all stations (bloom and non-bloom). It is most visible at a depth of ~40-60 m with a density of ~27.4-27.5 kg m^{-3} , which is shallower and less dense than the water mass where we see elevated iron associated with a clear deviation in T, S, and O₂, which is most pronounced at depths of 70-200 meters. Futhermore, this cold tongue of water is visible in our depth profiles at stations closer to the Ross Sea shelf and it is never associated with elevated iron concentrations.

The deviation in water mass properties between bloom and non-bloom stations that we demonstrate with density profiles in Figure 3 is also visible in the depth profiles (Figure S7) where we see a shift at bloom stations towards fresher, colder, and more oxygenated water between ~70-500 m. We can also see that the shift in temperature and oxygen is consistent with what we would expect from the turbulent entrainment of water from ~2000 m in these profiles. While this is not the exact water that we would expect to be turbulently entrained as we are ~150 km from KR1, it is the closest proxy that we have been able to find. We have also altered the text to reflect these changes (L462-477).

Salinity is less straightforward. While salinity does decrease with depth below ~300 m, it is not a huge change and cannot fully explain the salinity deviation observed at bloom stations relative to non-bloom stations. We have changed the text to reflect this. We have also added that the salinity deviation could be influenced by low salinity effluent from the vent, though

this is impossible to prove one way or the other without knowing the composition of the actual effluent.

We have also altered the text to present this argument as more of a hypothesis in need of further investigation.

Figure 1 and S3 needs the frontal zones labeled on the map. Given the number of references to them in the paper, the reader needs to see them on the figure without rereading the caption. You need a reference for maximum buoyancy frequency. Please give density range of UCDW as this would be important to figure S6. Might also indicate UCDW on figure S6

The contours on figure S6 indicate the isopycnals that define the boundaries of UCDW and are labeled with the density value for each isopycnal. We have added the density range for UCDW to the figure caption for figure S6 and to the manuscript L404.

The reference for maximum buoyancy frequency has been added to the Methods section.

The front labels have been added to Figure 1b, Figure S3b, and Figure S4b. We chose to add the labels to 1 panel in each figure and picked the panel where we thought the labels would be most visible.

Hydrothermal

Megaplumes (also known as event plumes) are formed episodically and thus are the antithesis of the source you want. You want long lived consistent sources. Invoking mega plumes is a mistake. There is no reason to include them in your discussion. Yes they have greater buoyancy but they are episodic and not seasonal.

The discussion of megaplumes has been removed.

Not all hydrothermal effluent is low salinity and hot fluids of normal and high salinity also rise into the water column. We know nothing about the end-member fluid composition at these sites, so I suggest that you omit this argument.

We never intended to say that the effluent from these vents was low salinity. We are aware that we know nothing about the end-member fluid composition at these sites and have been very careful to not make any assumptions about the effluent from these vents in our arguments. We have read through the section in question and cannot find a place where we make this statement. Without line numbers we cannot address this comment as we have been unable to identify the area of misunderstanding.

Referencing the German Treatise in Geo chemistry is pretty unfair to the people who did the work. To fully appreciate the dynamics of rising plumes would require a little more effort. For

example, high temperature fluids are diluted >1000X with ambient water within 10s of meters of the vent orifice. As a result, any plume that would affect the hydrography in the upper ocean would have the imprint of that deep water. Based on a wide deep body of literature there are many reasons to be careful about claiming a hydrographic anomaly from venting.

We have now cited the primary sources that were included in the German Treatise in Geochemistry.

Omitting the Mn data because you had trouble explaining it is a disappointing outcome.

These data were omitted because the other reviewers required them to be explained to the level that was not possible given our limited data and because they added little support to our argument.

Other points to consider

The data point at station 151, 1400m appears to be a mis-trip based on Fe, Nitrate, and Silicate.

This data point has been removed.

Figure 1A is not phytoplankton, per se, but chlorophyll.

Figure 1a is actually bathymetry. But we think we understand what you are getting at. We refer to the entire figure panel as showing bathymetry and phytoplankton distribution. We use the term phytoplankton distribution rather than Chlorophyll because Fig. 1b shows Net Primary Production and Fig. 1d shows Chlorophyll, but both are indicative of the distribution of phytoplankton. We feel that these terms are used appropriately in the figure caption.

Also, you should reach out to Andy Bowie about how he wants his data referenced. It seems really unfair to reference Schlitzer for his Fe and hydrography data. At least reference his cruise report. The data panel in ODV shows the data source.

The cruise identifier has been added to the caption of Figure S6. We have been unable to figure out how to cite the cruise report in any way that would give Dr. Bowie credit for the data that he collected.

I have looked at the revised manuscript and further address the hydrography question here. As currently described the hydrographic justifications are a weak part of this manuscript.

The idea that low salinity fluids hydrothermal fluids would result in the decrease in salinity seen here seems troublesome. Within meters of escaping the seafloor the vent fluids entrain ambient seawater and are diluted a 1000 fold within a meter or so and 5000 fold within 10s of meters and as you rise up further the dilution is even greater (See Lupton et al 1985?). So, for example let's take the approximate salinity of your deep water at 34.7 and let's go all out and suggest that the vent fluid has a salinity of 0 compared to the lowest known salinities from hydrothermal vents along the global ridge system of 10 or so. The decrease in salinity that you see is around 0.1 implying a dilution of vent fluid of 350. To complete this exercise, a 1000:1 dilution of 0 salinity endmember with 34.7 ambient seawater puts us at 34.665 and 5000:1 at 34.693 resulting in changes in salinity of .035 and 0.007. It would be tough to see these differences in the energetic upper ocean where strong gradients in physical properties exist. It is hard to do so in the deep ocean where the gradients are much smaller. The problem is compounded by Fe and Mn which have a range of concentrations. So let's conservatively say that vent fluid endmember values for phase separated fluids are 1000 μMol Fe and 300 μMol Mn. Here a 1000:1 dilution leaves 100nM and 30 nM of Fe and Mn respectively and a 5000:1 dilution 20 and 6nM of Fe and Mn respectively. These values would be very obvious in either the Fe or Mn data. You did not see them in your Mn data, which is why you MUST include the Mn data in the supplement. It is important to report these data. Based on your Fe values, I'm not sure I would expect Mn to change very much. I'd guess that the global mid-ocean ridge average of Fe to Mn in vent fluids is between 2 and 3. You seemingly have an Fe enrichment in the deep ocean of approximately 0.1nM Fe implying a Mn anomaly between 0.03nM and 0.05nM which is what your deep data show. In the surface, you have a source of Mn larger than that from hydrothermal. I'm not sure what it is and I have trouble invoking hydrothermal because the increase below the surface is observed throughout the data set. You don't need to explain Mn, but the data should be in the supplement or at least in the data set that you include when this paper gets accepted.

I went to the data set and I can now see that the deep water below the bloom site is more oxygen rich and colder than the water between 175-300 m. IN the manuscript you keep discussing UCDW (27.35-27.75) and the 27.6 to 27.8 density range when in reality the elevated Fe of interest lies between 150 and 300 m which is roughly in the range 27.7 to 27.8. Shallower than 27.7 the water becomes colder and fresher than the water at depth. That confused me. So for instance, you say on line 464 "..... are colder and more oxygenated than the overlying UCDW." And 466 ".... which is consistent with the properties of the anomalous water mass between the 27.6 to 27.8 kg m^{-3} isopycnals seen at bloom...." These statements are not entirely true if you look at the whole of the density ranges being discussed. Once you get above 150m or approx. 27.7 then your statements are no longer true.

Maybe you need to look at the hydrographic properties of the water above the ridge crest it instead of those in the deep water here which is probably Antarctic Bottom Water. The GIPY06 has data above the ridge crest and there is another set of data to the NE in the GLODAP data set. A quick glance says that it helps with salinity, but may hinder with respect to temperature. I previously asked Ed Baker if he has the CTD profiles from the Hahm et al manuscript and he does. So if you really want to lock this down, I suggest that you request the data from him.

Barring that, I think your best evidence for a ridge source of Fe is the “elevated” dFe observed down to >2000m and the particle release modeling that shows that hydrothermal plumes should move southeast. This may not be the water that is reaching the surface as that must have happened upstream of the bloom and downstream of the ridge crest where the deep water DFe values were likely higher (Referencing the paper from the ridge crest.)

We thank Reviewer 3 for their comments. Our response to each comment is in italics below the relevant comment.

I have looked at the revised manuscript and further address the hydrography question here. As currently described the hydrographic justifications are a weak part of this manuscript.

The idea that low salinity fluids hydrothermal fluids would result in the decrease in salinity seen here seems troublesome. Within meters of escaping the seafloor the vent fluids entrain ambient seawater and are diluted a 1000 fold within a meter or so and 5000 fold within 10s of meters and as you rise up further the dilution is even greater (See Lupton et al 1985?). So, for example let's take the approximate salinity of your deep water at 34.7 and lets go all out and suggest that the vent fluid has a salinity of 0 compared to the lowest known salinities from hydrothermal vents along the global ridge system of 10 or so. The decrease in salinity that you see is around 0.1 implying a dilution of vent fluid of 350. To complete this exercise, a 1000:1 dilution of 0 salinity endmember with 34.7 ambient seawater puts us at 34.665 and 5000:1 at 34.693 resulting in changes in salinity of .035 and 0.007. It would be tough to see this differences in the energetic upper ocean where strong gradients in physical properties exist. It is hard to do so in the deep ocean where the gradients are much smaller. The problem is compounded by Fe and Mn which have a range of concentrations. So let's conservatively say that vent fluid endmember values for phase separated fluids are 1000 uMol Fe and 300 uMol Mn. Here a 1000:1 dilution leaves 100nM and 30 nM of Fe and Mn respectively and a 5000:1 dilution 20 and 6nM of Fe and Mn respectively. These values would be very obvious in either the Fe or Mn data. You did not see them in your Mn data, which is why you MUST include the Mn data in the supplement. It is important to report these data. Based on your Fe values, I'm not sure I would expect Mn to change very much. I'd guess that the global mid-ocean ridge average of Fe to Mn in vent fluids is between 2 and 3. You seemingly have an Fe enrichment in the deep ocean of approximately 0.1nM Fe implying a Mn anomaly between 0.03nM and 0.05nM which is what your deep data show. In the surface, you have a source of Mn larger than that from hydrothermal. I'm not sure what it is and I have trouble invoking hydrothermal because the increase below the surface is observed throughout the data set. You don't need to explain Mn, but the data should be in the supplement or at least in the data set that you include when this paper gets accepted.

- ***We have altered the language to remove any mention of low-salinity hydrothermal vent fluids. (L475)***
- ***The Mn data is available with the rest of the data associated with this paper***

I went to the data set and I can now see that the deep water below the bloom site is more oxygen rich and colder than the water between 175-300 m. IN the manuscript you keep discussing UCDW (27.35-27.75) and the 27.6 to 27.8 density range when in reality the elevated Fe of interest lies between 150 and 300 m which is roughly in the range 27.7 to 27.8. Shallower than 27.7 the water becomes colder and fresher than the water at depth. That confused me. So for instance, you say on line464 "... are colder and more oxygenated than the overlying UCDW." And 466 "... which is consistent with the properties of the anomalous water mass between the 27.6 to 27.8 kg m⁻³ isopycnals seen at bloom..." These statements are not entirely true if you look at the whole of the density ranges being discussed. Once you get above 150m or approx. 27.7 then your statements are no longer true. Maybe you need to look at the hydrographic properties of the water above the ridge crest it instead of those in the deep water here which is probably Antarctic Bottom Water. The GIPY06 has data above the ridge crest and there is another set of data to the NE in the GLODAP data set. A quick glance says that it helps with salinity, but may hinder with respect to temperature. I previously asked Ed Baker if he has the CTD profiles from the Hahm et al manuscript and he does. So if you really want to lock this down, I suggest that you request the data from him.

- ***We have changed the language to make it clearer that the elevated DFe concentrations are associated with an anomalous water mass that spans the 27.6 to 27.8 density range. L455***

- *The reviewer is correct that shallower than 27.7 the water becomes colder and fresher than the water at depth. We are not only comparing the water between 27.6-27.8 to water at depth we are also comparing it to water at the same density range at non-bloom stations. Our point is that at bloom stations water in the density range from 27.6-27.8 is fresher, colder, and more oxygenated than at non-bloom stations, and that there is a clear deviation from the linear mixing line. At non-bloom stations we see a linear mixing line between the deeper warm, salty, low-oxygen waters at 27.8 and the cold, fresh, more oxygenated waters at 27.6. At bloom stations there is a distinct deviation from the linear mixing relationship caused by the mixing in of an additional water mass that is pulling the linear mixing relationship in a colder, fresher, and more oxygenated direction.*
- *We have reached out to Ed Baker for the relevant CTD data from the Hahn et al. (2015) manuscript but have not received the data at this time.*

Barring that, I think your best evidence for a ridge source of Fe is the “elevated” dFe observed down to >2000m and the particle release modeling that shows that hydrothermal plumes should move southeast. This may not be the water that is reaching the surface as that must have happened upstream of the bloom and downstream of the ridge crest where the deep water DFe values were likely higher (Referencing the paper from the ridge crest.)

- *We agree with what the reviewer is stating in this paragraph.*